JCB Journal of Cell Biology

## REPORT

# A novel function for the sperm adhesion protein IZUMO1 in cell–cell fusion

Nicolas G. Brukman[1]*![ORCID], Kohdai P. Nakajima[2]*, Clari Valansi[1], Kateryna Flyak[1]![ORCID], Xiaohui Li[1], Tetsuya Higashiyama[2,3,4], and Benjamin Podbilewicz[1]![ORCID]

**Mammalian sperm–egg adhesion depends on the trans-interaction between the sperm-specific type I glycoprotein IZUMO1 and its oocyte-specific GPI-anchored receptor JUNO. However, the mechanisms and proteins (fusogens) that mediate the following step of gamete fusion remain unknown. Using live imaging and content mixing assays in a heterologous system and structure-guided mutagenesis, we unveil an unexpected function for IZUMO1 in cell-to-cell fusion. We show that IZUMO1 alone is sufficient to induce fusion, and that this ability is retained in a mutant unable to bind JUNO. On the other hand, a triple mutation in exposed aromatic residues prevents this fusogenic activity without impairing JUNO interaction. Our findings suggest a second function for IZUMO1 as a unilateral mouse gamete fusogen.**

## Introduction

The final steps of mammalian egg–sperm fusion remain a mechanistic enigma. Cell fusion requires the action of specialized proteins, named fusogens, to overcome the energetic barriers that arise when two plasma membranes come into proximity (Chernomordik and Kozlov, 2003). Authentic fusogens are both necessary in their system of origin and sufficient to induce membrane merging in otherwise nonfusing heterologous systems (Segev et al., 2018). While a mystery in mammals, gamete fusion in flowering plants and protists is mediated by the fusogen generative cell-specific 1/hapless 2 (GCS1/HAP2). The essentiality of GCS1 in gamete fusion was demonstrated in *Arabidopsis thaliana*, being sperm-expressed and necessary for gamete fusion (von Besser et al., 2006; Mori et al., 2006; Johnson et al., 2004). GCS1 is also essential to fuse gametes in the malaria parasite *Plasmodium*, in the slime mold *Dictyostelium*, and in the algae *Chlamydomonas* (Liu et al., 2008; Hirai et al., 2008; Okamoto et al., 2016). It was subsequently demonstrated that the expression of *A. thaliana* and *Plasmodium falciparum* GCS1 is sufficient to fuse mammalian cells in culture (Valansi et al., 2017; Kumar et al., 2022); thereby GCS1 is a bona fide fusogen. GCS1 structure is similar to class II viral glycoproteins (e.g., rubella and zika viruses; Fédry et al., 2017; Pinello et al., 2017; Valansi et al., 2017; Feng et al., 2022) and fusion family (FF) proteins from nematodes and other organisms (Mohler et al., 2002; Sapir et al., 2007; Pérez-Vargas et al., 2014). This protein superfamily, termed Fusexins (Valansi et al., 2017), are widely distributed in multiple eukaryotic and archaeal phyla (Moi et al., 2022), but to

date no members have been identified in vertebrates (Brukman et al., 2022).

Unlike gamete fusogens, mammalian somatic fusogens are known. For example, fusion of myoblasts requires Myomaker (TMEM8c) and Myomerger (Myomixer/Minion/Gm7325; Millay et al., 2013; Bi et al., 2017; Quinn et al., 2017; Zhang et al., 2017). Their expression in fibroblasts drives cell fusion: Myomerger can work unilaterally from either one of the merging membranes, while Myomaker is required on both fusing cells (Leikina et al., 2018). During placenta formation, trophoblast fusion is mediated by syncytins (Lavialle et al., 2013). *Syncytin-A* and *-B* mutations in mice result in fusion defects during the formation of the syncytiotrophoblast (Dupressoir et al., 2011; Dupressoir et al., 2009), while human Syncytin-1 or -2 expression is sufficient to induce cell fusion (Esnault et al., 2008; Blond et al., 2000).

Before gamete fusion, the sperm must undergo capacitation, which includes the exocytosis of the acrosome, a specialized vesicle in the head (Yanagimachi, 1994; Visconti et al., 2011). This allows the sperm to penetrate a proteinic coating that covers the egg, called *zona pellucida* (ZP) in mammals (Wassarman, 1999). Only after penetration of the ZP, the plasma membranes of both gametes can bind to each other and finally fuse (Bianchi and Wright, 2020). Some proteins expressed in the gametes are essential for the last steps of fertilization (Deneke and Pauli, 2021). The oocyte tetraspanins CD9 and CD81 are required for gamete fusion (Kaji et al., 2000; Le Naour et al., 2000; Miyado et al., 2000; Rubinstein et al., 2006) by regulating

[1]Department of Biology, Technion - Israel Institute of Technology, Haifa, Israel;   [2]Division of Biological Science, Graduate School of Science, Nagoya University, Nagoya, Aichi, Japan;   [3]Institute of Transformative Bio-Molecules, Nagoya University, Nagoya, Aichi, Japan;   [4]Department of Biological Sciences, Graduate School of Science, University of Tokyo, Tokyo, Japan.

*N.G. Brukman and K.P. Nakajima contributed equally to this paper.   Correspondence to Benjamin Podbilewicz: podbilew@technion.ac.il;   Tetsuya Higashiyama: higashi@bs.s.u-tokyo.ac.jp;   Nicolas G. Brukman: nbrukman@gmail.com.



membrane architecture (Runge et al., 2007; Inoue et al., 2020). Mutation of any of the sperm-specific proteins TMEM95, SPACA6, FIMP, SOF1, and DCST1/2 leads to male infertility (Barbaux et al., 2020; Noda et al., 2020; Lorenzetti et al., 2014; Lamas-Toranzo et al., 2020; Fujihara et al., 2020; Inoue et al., 2021; Noda et al., 2022). While these genes are essential for late stages in fertilization, and loss-of-function mutations of any of them prevent gamete fusion, it is not clear what specific step in the process is affected. The only known pair of trans-interacting proteins is IZUMO1, in the sperm, and JUNO/IZUMO1R, from the egg. IZUMO1 deletion blocks gamete fusion (Inoue et al., 2005), and JUNO was identified as the IZUMO1 receptor in the oocyte (Bianchi et al., 2014). Evidence from infertile patients suggests that these proteins play a role during human fertilization (Clark and Naz, 2013; Yu et al., 2018; Enoiu et al., 2022). The IZUMO1–JUNO interaction mediates gamete binding (Matsumura et al., 2021) in a species-specific manner (Bianchi and Wright, 2015). WT eggs can bind to cells expressing IZUMO1 in a JUNO-dependent manner (Inoue et al., 2013, 2015; Chalbi et al., 2014; Ohto et al., 2016). In contrast, sperm can fuse to somatic cells (Rival et al., 2019; Mattioli et al., 2009; Bendich et al., 1974). However, a previous study failed to detect fusion between mixed somatic cells expressing JUNO or IZUMO1 (Bianchi et al., 2014). Here, we present evidence that the expression of IZUMO1 is sufficient to mediate cell fusion independently of its binding to JUNO, thus underpinning an additional role for IZUMO1 in membrane merger during fertilization.

## Results and discussion

### IZUMO1 is sufficient to fuse cells

There are two attributes that define a fusogen: First, the molecule must be necessary for membrane fusion so when it is absent or mutated fusion fails; and second, the protein must be sufficient to merge cells that normally do not fuse. Generally nonfusing cells in culture, such as baby hamster kidney (BHK) cells, have been widely used to test the fusogenic ability of proteins and illustrate fusogen sufficiency (Avinoam et al., 2011; Valansi et al., 2017; White et al., 1981; Waning et al., 2004; Jeetendra et al., 2002; Bitto et al., 2016). To look for gamete fusogen(s) in mammals, we aimed to evaluate candidate gamete-specific transmembrane proteins using this system, with mouse egg JUNO and sperm IZUMO1 as negative controls to account for trans-interacting proteins involved in binding but not fusion (Fig. 1). To our surprise, when we expressed IZUMO1, but not JUNO, we observed syncytia formation at levels like those obtained with GCS1, a sperm fusogen from *Arabidopsis* (Fig. 1, A and B). We found that multinucleation increased threefold when IZUMO1 was expressed compared to the negative control containing a myristoylated EGFP (myrGFP; Fig. 1 B). For both IZUMO1 and GCS1, we found an increase in cells with two to five nuclei (Table S1). The presence of both IZUMO1 and JUNO, on the surface of BHK cells, was confirmed by immunostaining of nonpermeabilized cells (Fig. S1, A and B). To determine whether IZUMO1-induced syncytia formation was due to failure in cytokinesis, we incubated BHK cells with and without an inhibitor of the cell cycle (5-fluoro-2′-deoxyuridine [FdUrd]; Valansi et al., 2017). We found the cells failed to proliferate but there was no

reduction in multinucleation, suggesting that IZUMO1-mediated multinucleation is not due to failure in cytokinesis (Fig. S1 D).

We then asked whether IZUMO1 and JUNO can mediate cell fusion using content mixing experiments (Avinoam et al., 2011). In this assay, two populations of BHK cells expressing either cytosolic EGFP (GFPnes) or nuclear H2B-RFP are co-incubated, and the appearance of multinucleated cells containing both fluorescent markers indicates cell fusion (Fig. 2). We found that expression of JUNO failed to induce content mixing. In contrast, IZUMO1 induced a 5.5-fold increase in the content mixing of BHK cells compared to the controls employing the fluorescent empty vectors, while a fourfold increase was observed in the positive GCS1 control (Fig. 2, A, B, and D; and Fig. S2 A). To determine whether IZUMO1 can merge different cells, we performed content mixing assays using human HEK293T cells and obtained similar results (Fig. S2, B and C). Thus, the mouse IZUMO1 is sufficient to fuse hamster and human cells.

### IZUMO1-mediated fusion uses a viral-like mechanism

Viral fusogens, like the hemagglutinin of influenza virus (White et al., 1982), FAST proteins of reoviruses (Shmulevitz and Duncan, 2000) or the virus-related syncytins (Esnault et al., 2008), can mediate fusion when they are expressed in one of the fusing membranes while other fusogens, like *Caenorhabditis elegans*' EFF-1 and AFF-1, are required in both fusing membranes (Podbilewicz, 2014). As IZUMO1 is present only in the sperm, we asked whether cells expressing IZUMO1 can fuse to cells transfected with an empty vector or expressing JUNO. We found that content mixing was observed when IZUMO1 was present only in one of the two populations of cells that were mixed, implying a unilateral fusion mechanism like in viruses (Fig. 2 and Fig. S2 A). Expressing JUNO in trans of IZUMO1 appeared to increase the number of fusion events, which may be due to increased association between cells from the two populations mediated by IZUMO1–JUNO interaction. The unilateral activity of IZUMO1 is further supported by the fact that mouse sperm can fuse to BHK cells expressing JUNO and not to control BHK cells (Fig. S3, A and B). Thus, our results suggest that mouse gamete fusion can be mediated by the unilateral fusogenic activity of IZUMO1 on the sperm that follows the docking mediated by the IZUMO1–JUNO interaction (Fig. S3 E).

### Dynamics of IZUMO1-mediated fusion

To independently study the apparent fusogenic activity of IZUMO1, we performed live imaging experiments to track fusion events of BHK cells (Fig. 3). We transfected cells with a plasmid encoding for cytoplasmic RFP (RFPcyto) alone, for the negative control, or together with a mifepristone-inducible system to express our candidate proteins. As a positive control, we used EFF-1, a somatic fusogen from *C. elegans* (Avinoam et al., 2011). Vectors for E-cadherin (Truffi et al., 2014), GEX2 (gamete expressed 2), IZUMO1, and JUNO were similarly prepared, and the cells were visualized upon induction. We observed that cell fusion occurred following EFF-1 expression (Fig. 3 and Video 2). Cell fusion events were quantified 12 h after induction and typically occur between sister cells several hours after cytokinesis. Sister cells can be more likely to fuse due to their close contact and the fact that they are in the same stage of the cell

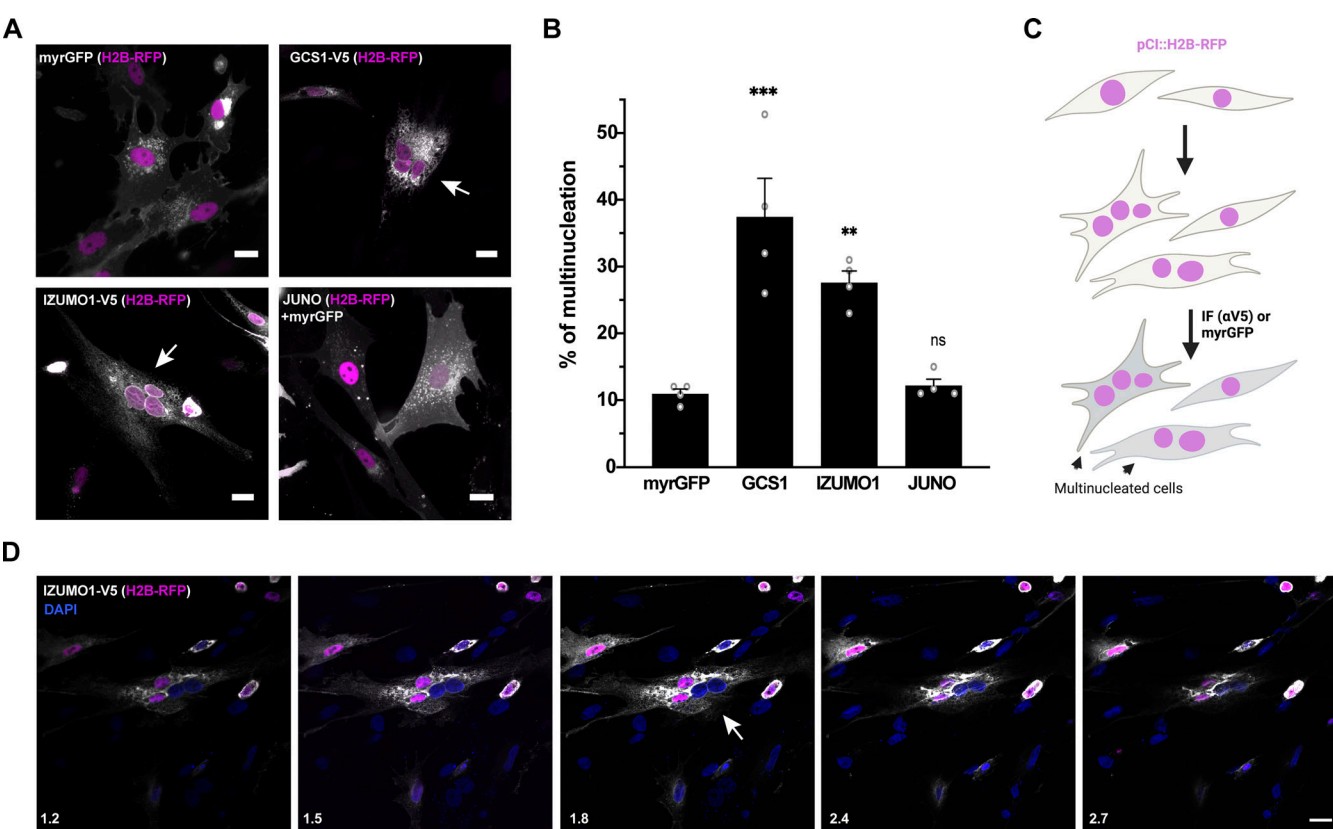

Figure 1. **IZUMO1 induces multinucleation of BHK cells. (A)** Cells were transfected with either pCI::myrGFP::H2B-RFP (myrGFP), pCI::GCS1/HAP-V5::H2B-RFP, pCI::IZUMO1-V5:H2B-RFP, or pCI::JUNO::H2B-RFP vectors. JUNO was co-transfected with a plasmid for myrGFP. Immunofluorescence was performed with anti-V5 antibodies for IZUMO1 and GCS1/HAP2, while the myrGFP signal is shown for JUNO and vector control (gray). Arrows show cells with more than one nucleus (magenta). Scale bars, 20 μm. **(B)** The percentage of multinucleation was defined as the ratio between the nuclei in multinucleated cells (NuM) and the total number of nuclei in multinucleated and fluorescent cells in contact that did not fuse (NuC), as follows: % of multinucleation = (NuM/[NuM + NuC]) × 100 (see Table S1). We show individual data and means ± SEM of four independent experiments. The total number of nuclei counted was: myrGFP, 3,846; GCS1/HAP2, 3,301; IZUMO1, 3,353; JUNO, 3,873. The distribution of the different mononucleated and multinucleated cells counted can be found in Table S1. Comparisons were made with one-way ANOVA followed by Dunett's test against the empty vector. **P < 0.01, ***P < 0.001. **(C)** Scheme of experimental design. **(D)** Z-Series of a tetranucleate IZUMO-V5 cell (white arrow). After immunofluorescence using anti-V5 antibody, the syncytium was analyzed by spinning disk confocal microscopy. DAPI-stained nuclei (blue), IZUMO1-V5 (gray), and H2B-RFP (magenta) are shown. The intensity of the nuclear marker differed between nuclei (white arrow), suggesting a recent fusion between cells expressing different levels of H2B-RFP. Numbers on the bottom left side of each panel are optical sections in micrometers. Scale bar, 20 μm. See also Fig. S1 C and Video 1.

cycle. We found that EFF-1 expression increased fusion levels (Fig. 3 B), whereas the adhesion proteins (E-cadherin, GEX2, and JUNO) did not display fusogenic activity; significantly, IZUMO1 induced cell fusion (Fig. 3 B and Videos 3, 4, 5, and 6). Because IZUMO1 was expressed fused to Venus in the C terminus, we could follow its expression; we often observed Venus fluorescence at time points when fusion started taking place (Fig. 3 A and Video 3). Taken together, our content mixing and live imaging results support a model in which IZUMO1 acts independently of JUNO in a unilateral mechanism to fuse mammalian cells (Fig. S3 E).

### Cells expressing IZUMO1 attach to mouse eggs but fail to fuse

To study a semi-heterologous system, which retains the endogenous components of the egg, but not sperm, we incubated BHK cells expressing IZUMO1 along with the nuclear marker H2B-RFP with mouse eggs expressing the tetraspanin protein CD9-GFP, and fusion was evaluated (Fig. S3 C). We found that IZUMO1 expression was sufficient to mediate the binding of

BHK cells to eggs but did not induce their fusion (Fig. S3 D), consistent with previous reports (Inoue et al., 2013; Inoue et al., 2015; Chalbi et al., 2014; Ohto et al., 2016). As a positive control, we used a viral fusogen, the vesicular stomatitis virus G protein (VSV-G; Florkiewicz and Rose, 1984). We co-expressed IZUMO1 together with VSV-G to mediate BHK attachment to the eggs and lowered the pH to trigger VSV-G activity. Only in this condition, we observed chromosomes from the BHK in the cytoplasm of the eggs (Fig. S3 D). Thus, while viral VSV-G can fuse BHK cells to eggs, we could not demonstrate a similar role for IZUMO1 under the conditions employed, which might suggest a role for additional cofactors required to trigger fusion or bypass the mechanisms that block fusion to the egg.

### Dissection of the JUNO-binding and membrane fusion activities of IZUMO1

Some viral fusogens accomplish membrane docking and merger using different structural domains (White et al., 2008). We hypothesized that IZUMO1 will similarly have different domains

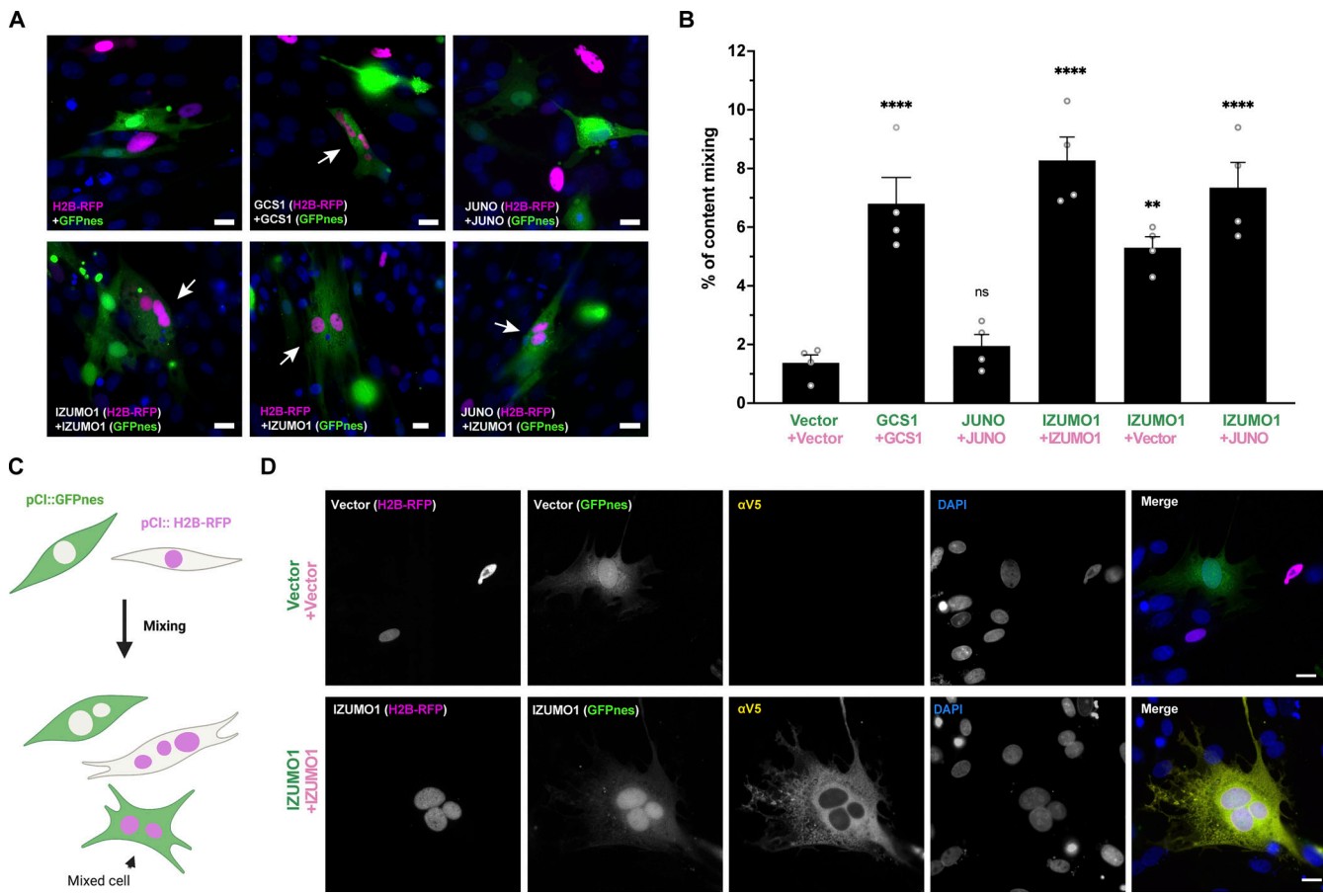

Figure 2. **IZUMO1 induces fusion of BHK cells. (A)** Representative images of mixed cells transfected with pCl::GFPnes or pCl::H2B-RFP empty vectors or containing the coding sequence for the expression of GCS1/HAP2, IZUMO1, and JUNO as indicated. Arrows show content mixed cells containing both GFPnes (green cytoplasm) and H2B-RFP (magenta nuclei). DAPI staining is shown in blue. Scale bars, 20 µm. See also Fig. S2 A. **(B)** Quantification of content mixing experiments. The percentage of mixing was defined as the ratio between the nuclei in mixed cells (NuM) and the total number of nuclei in mixed cells and fluorescent cells in contact that did not fuse (NuC), as follows: % of mixing = (NuM/[NuM + NuC]) × 100. Bar chart showing individual experiment values (each corresponding to 1,000 nuclei) and means ± SEM of four independent experiments. Comparisons by one-way ANOVA followed by Dunett's test against the empty vectors. **$P < 0.01$, ****$P < 0.0001$. **(C)** Scheme of experimental design. **(D)** Representative images of cells transfected with empty or IZUMO1-coding vectors after a content mixing assay and subjected to immunostaining using an anti-V5 antibody. Each separate channel (red, green, far red, and DAPI) and merge are shown. Scale bars, 20 µm.

mediating its dual activities. To study the structural features involved in IZUMO1 fusogenic activity, we generated a series of mutants based on the crystal structure of IZUMO1 (Nishimura et al., 2016, PDB: 5B5K) and its interactions with the docking receptor JUNO (Fig. 4, A–C). To assess the functionality of these mutant proteins, we studied their localization by surface biotinylation (Fig. 4 D) and immunostaining (Fig. 4 E) and compared their activities in binding and content-mixing experiments. IZUMO1 contains a domain composed of a four-helix bundle in the N-terminal part (Izumo domain) and an Ig-like domain linked by a β-hairpin or hinge (Fig. 4 A). Members of the Fusexin superfamily and a few other single-pass transmembrane proteins contain Ig-like domains and have been involved in cell fusion (Vance and Lee, 2020; Martens and McMahon, 2008). We found that the deletion of the transmembrane domain produced an inactive form of the protein (IZUMO[Ecto]; Fig. 5 A) that was not detected on the surface of the cells, suggesting that anchoring to the plasma membrane is

required for fusion. When the Ig-like domain of IZUMO1 was absent, the protein IZUMO1[ΔIg] was expressed intracellularly and at low levels on the plasma membrane as determined by surface biotinylation (Fig. 4 D) but not detected on the cellular surface by immunofluorescence on non-permeabilized cells (Fig. 4 E). While detected at low levels on the surface at steady state, IZUMO1[ΔIg] was able to induce content-mixing compared to the empty vector, although lower than WT IZUMO1 (Fig. 5 A). These results suggest that IZUMO1[ΔIg] was able to transiently reach the cell surface at low levels and was able to induce fusion. This change in the localization could be explained by the lack of the single glycosylation site at the Ig-like domain that protects the protein from degradation (Inoue et al., 2008).

Thus, IZUMO[Ecto] cannot mediate BHK cell fusion, suggesting that the transmembrane domain is necessary for fusion, as is the case for viral fusogens (Armstrong et al., 2000) and the eukaryotic fusogen EFF-1 (Podbilewicz et al., 2006). In contrast, the Ig-like domain is required for correct trafficking and localization to

**Figure 3. Quantification of fusion activity using live imaging. (A)** Time-lapse images from a fusion assay. BHK cells were transfected with plasmids for expression of RFPcyto (magenta) and EFF-1 (co-transfection) or IZUMO1-Venus (green). Arrowheads and arrows indicate contacting and fused cells, respectively. Time (h:min) after the start of observation (see Videos 2, 3, 4, 5, and 6). Sister cells generally fuse several hours after they divide, suggesting that the probability of fusion increases when the cells are at the same stage of the cell cycle. Scale bars, 50 µm. **(B)** Quantification of live imaging experiments in which BHK cells express RFPcyto, E-cadherin, EFF-1, GEX2, IZUMO1-Venus, or JUNO. The percentage of fusion was defined as the ratio between the number of fusion events (Fe) and the number of transfected cells (Tc), as follows: % of fusion events = (Fe/Tc) × 100. Bar chart showing individual experiments and means ± SEM of at least three independent experiments. Total Tc analyzed = 1,001 (RFP), 1,179 (E-cadherin), 930 (EFF-1), 415 (GEX2), 817 (IZUMO1), and 1,265 (JUNO). Comparisons by one-way ANOVA followed by Dunett's test against RFPcyto. **P < 0.01. **(C)** Scheme of experimental design.

the plasma membrane but is not essential for fusion, since IZUMO1$^{\Delta Ig}$ can mediate BHK cell fusion.

To study whether the fusogenic activity of IZUMO1 is correlated with the ability to bind JUNO, we analyzed the W148A mutant that is essential for IZUMO1–JUNO interaction (Fig. 4 A; Aydin et al., 2016; Ohto et al., 2016). We found that IZUMO1$^{W148A}$ had no effect on the levels of content mixing compared to WT IZUMO1 (Fig. 5 A) but disrupted the binding of BHK cells to eggs (Fig. 5, B and C). Unlike IZUMO$^{Ecto}$ or IZUMO1$^{\Delta Ig}$, the W148A mutation did not affect its localization on the surface (Fig. 4, D and E). Thus, the W148 residue on the β-hairpin of IZUMO1 is required for BHK binding the egg through JUNO but does not play a function in BHK cell fusion.

The four-helix bundle of the Izumo domain contains three exposed aromatic residues (Fig. 4, A and B; and Video 7), which

do not form any obvious fusion loop or hydrophobic patch but may be important for the interaction of IZUMO1 to the membrane of the egg or to other proteins given that JUNO is displaced from the fusion site shortly after the apposition of the fusing membranes (Inoue et al., 2015). To test whether these residues are required for fusion, we generated a triple mutant F28A, W88A and W113A (FWW, Fig. 4 C). IZUMO1$^{FWW}$ was detected on the surface of BHK cells (Fig. 4, D and E) and was able to mediate BHK binding to the egg (Fig. 5, B and C); however, this mutant induced lower levels of content mixing of BHK cells than WT IZUMO1 (Fig. 5 A). Thus, our results support the existence of two functional domains in IZUMO1, with binding to JUNO mediated through W148 on the hinge region and cell fusion dependent on three exposed aromatic residues on the four-helix bundle of the Izumo domain.

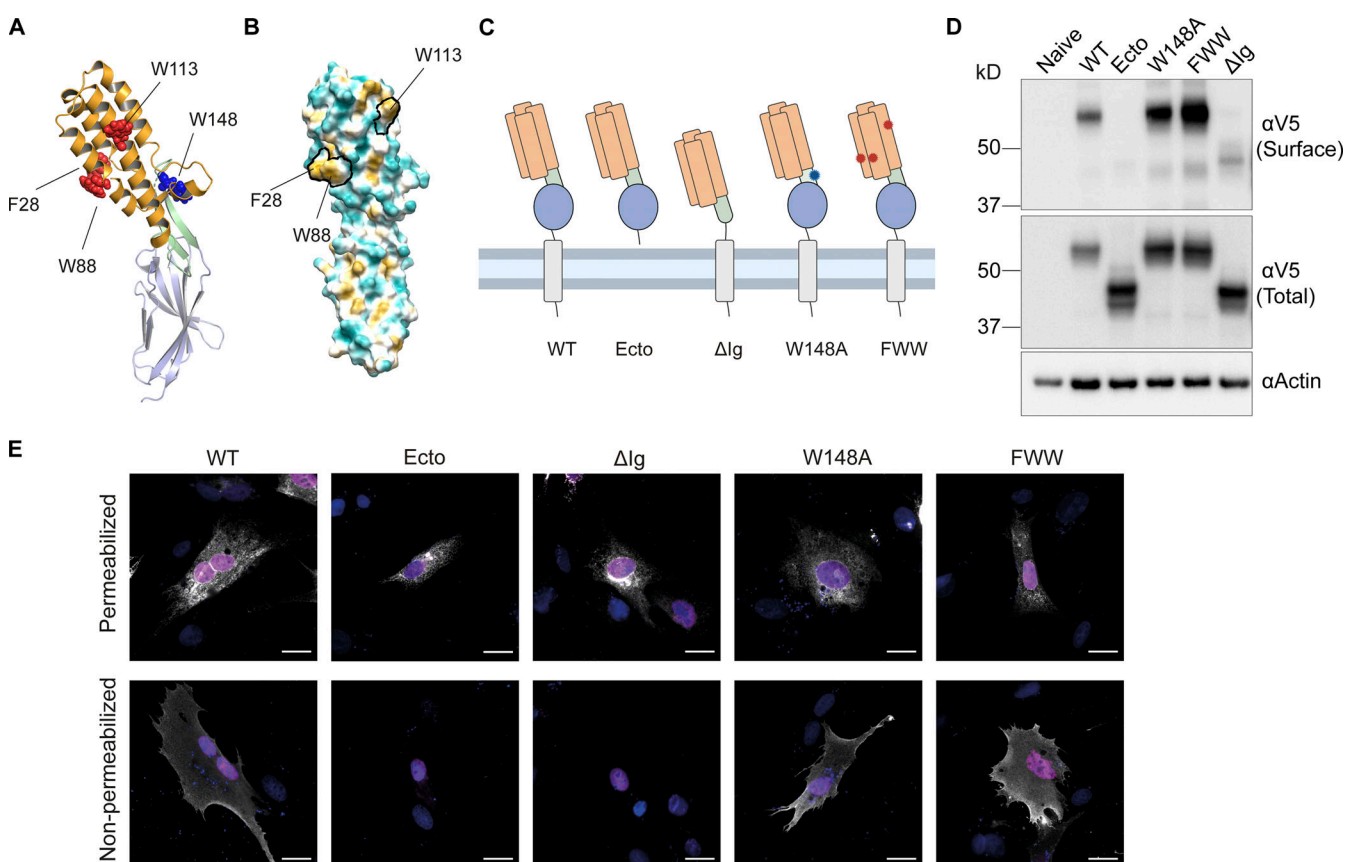

Figure 4. **Mutagenesis of IZUMO1 and expression of mutant proteins. (A)** Structure of IZUMO1 (PDB: 5B5K [Nishimura et al., 2016]) showing the four-helix bundle (in orange) containing four solvent-exposed aromatic residues (in red, F28, W88, and W113), the hinge (in light green) with the JUNO-interacting W148 (in blue), and the Ig-like domain (in teal). **(B)** Structure of IZUMO1 (PDB: 5B5K [Nishimura et al., 2016]) showing solvent-excluded surface colored based on molecular lipophilicity potential maps, ranging from dark cyan (hydrophilic) to dark gold (lipophilic). The aromatic residues F28, W88, and W113 are contoured in black. **(C)** Schematic representation of WT IZUMO1 and the mutants, maintaining the color coding for the different domains as in A. The mutants with the deletion of the transmembrane domain and cytoplasmic tail (Ecto), the deletion of the Ig-like domain (ΔIg), the point mutation W148A, and the triple mutant (FWW) are represented. **(D)** Representative Western blot of naive BHK cells or transfected with a plasmid encoding for WT IZUMO1, or the mutants shown in B. The different variants were detected with an anti-V5 antibody. "Surface" indicates surface biotinylation followed by affinity purification using neutravidin agarose beads; "Total" indicates the expression in whole cell extracts. Actin is used as a loading control. **(E)** Representative images of cells transfected with the pCI::H2B-RFP vectors encoding for WT IZUMO1 or the different mutants and subjected to an immunostaining (in white) after permeabilization with Triton X-100 using an anti-V5 antibody or without permeabilization using an antibody against the Izumo domain (Mab120). The signals for H2B-RFP and DAPI are shown in magenta and blue, respectively. The proportion of non-permeabilized cells showing surface expression was: IZUMO1$^{WT}$ (41.9%, $n = 1,013$), IZUMO1$^{ΔIg}$ (0%, $n = 1,000$), IZUMO1$^{Ecto}$ (0%, $n = 1,000$), IZUMO1$^{W148A}$ (60.3%, $n = 1,005$), IZUMO1$^{FWW}$ (61.7%, $n = 995$). Scale bars, 20 μm. Source data are available for this figure: SourceData F4.

### Aromatic residues on the four-helix bundle are required for fusion independently of the JUNO–IZUMO1 interactions

There is no homology between IZUMO1 and any known fusogen. IZUMO1 lacks a hydrophobic stretch that could work as a fusion peptide or loop, which serves to anchor to the opposing membrane (Aydin et al., 2016), a characteristic feature shared by many, but not all, cellular and viral fusogens (Brukman et al., 2019). The four-helix bundle of the Izumo domain contains solvent-exposed aromatic residues (Nishimura et al., 2016), which are not involved in JUNO–IZUMO1 interaction (Aydin et al., 2016; Ohto et al., 2016) but may interact with the egg. This is supported by our findings that IZUMO1$^{FWW}$ had normal JUNO-mediated binding of BHK cells to eggs, and a lower fusogenic activity in BHK cell fusion (Fig. 5). While F28 is well conserved in rodents, W88 and W113 are present in most IZUMO1 mammalian orthologs (Ohto et al., 2016; Aydin et al.,

2016). Interestingly, antibodies against the N-terminal region of IZUMO1 inhibited gamete fusion in vitro without altering sperm binding to the egg (Inoue et al., 2013), supporting an additional role of IZUMO1 in fusion besides binding. Alternatively, the aromatic residues on the surface of the Izumo domain may be required for the interaction with an unknown secondary receptor (Inoue et al., 2015), for oligomerization (Ellerman et al., 2009) or for interaction with other domains within IZUMO1 after a conformational change (Inoue et al., 2015).

Following sperm–egg binding, JUNO is excluded from the interface between gametes, while IZUMO1 is conversely enriched in this region (Inoue et al., 2015). IZUMO1 concentrates at the fusion zone where it dimerizes and undergoes a conformational change mediated by a disulfide isomerase (Inoue et al., 2015). Furthermore, IZUMO1 has a more ancestral origin, being found in many vertebrate phyla, while

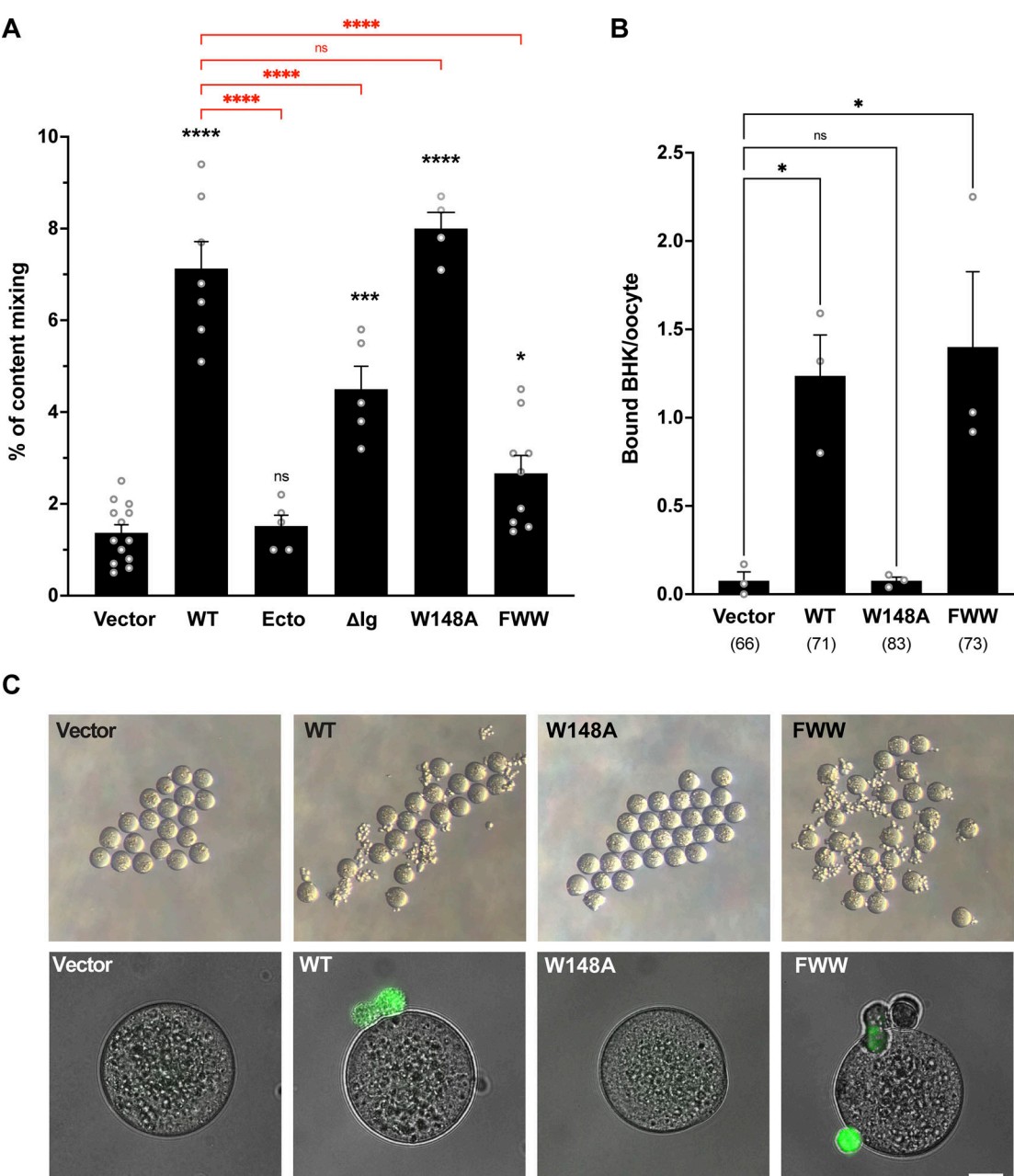

**Figure 5. Functional characterization of IZUMO1 mutants. (A)** Quantification of content mixing in BHK cells expressing vectors, WT IZUMO1, and its mutants (Ecto, ΔIg, W148A, and FWW; see Fig. 4). Bar chart showing means ± SEM. $n$ = 1,000 nuclei per independent experiment. Comparisons by one-way ANOVA followed by Bonferroni's test against the vector (black) and against WT (red). *P < 0.05, ***P < 0.001, ****P < 0.0001. **(B and C)** BHK-oocyte binding assay. BHK cells expressing IZUMO1 (WT and mutants). **(B)** Quantification of the binding of BHK cells to oocytes. Cells were transfected with pCI::GFPnes empty vector or encoding for IZUMO1, IZUMO1$^{W148A}$, or IZUMO1$^{FWW}$ and incubated with WT oocytes. The number of BHK cells bound per oocyte was determined; $n$ = total number of oocytes analyzed in parenthesis. Bar chart showing means ± SEM. Comparisons by one-way ANOVA followed by Dunnett's test against the vector. *P < 0.05. **(C)** Representative images of oocytes from one experiment in B taken under a dissecting microscope (upper row) and a wide-field illumination microscope showing the merged differential interference contrast (DIC) and GFP channels (lower row). Scale bar, 20 µm.

JUNO is found in mammals only (Grayson, 2015). This evolutionary distribution and IZUMO1's unilateral fusogenic activity suggest a function for IZUMO1 in fertilization that is independent of its known interaction with JUNO. This is further supported by our finding that IZUMO1$^{W148A}$ which fails to bind JUNO can induce cell fusion, and that IZUMO1$^{FWW}$ can mediate BHK-egg binding while BHK–BHK cell fusion was reduced.

**Are other gamete membrane proteins involved in fusion?**
While our results point to IZUMO1 as a bona fide fusogen that can fuse hamster and human cells in culture, this does not exclude the possibility that other fusogenic proteins work cooperatively with IZUMO1 to ensure efficient fertilization in vivo (Inoue et al., 2013; Inoue et al., 2015; Chalbi et al., 2014; Gaikwad et al., 2019). This kind of cooperation exists in plant fertilization, where the mutation of DMP8/9 proteins drastically reduces the

activity of GCS1 (Cyprys et al., 2019; Zhang et al., 2019). Proteins related to IZUMO1, for instance, IZUMO2-4, SPACA6, or TMEM95, may work as fusogens or potentiate IZUMO1's fusogenic activity (Deneke and Pauli, 2021). In this sense, IZUMO1 can form multiprotein complexes (Contreras et al., 2022; Ellerman et al., 2009; Gaikwad et al., 2019). The absence of these proteins may explain the inability of somatic cells expressing IZUMO1 to fuse to eggs (Fig. S3 D). Other factors, such as the movement of the flagella (Ravaux et al., 2016) or the exposure of phosphatidylserine on the sperm (Rival et al., 2019), may also be required for gamete fusion. Another factor that may explain the ability of IZUMO1 to mediate somatic cell fusion but not between BHK cells and eggs is the presence of nanotubes in the cell lines used. In this scenario, IZUMO1 can drive the efficient formation and expansion of the nanotubes yielding multinucleated cells allowing efficient content mixing.

Additionally, the fact that other eukaryotic fusogens (e.g., EFF-1, AFF-1, and GCS1) were shown to be less efficient in vitro than their viral counterparts (Avinoam et al., 2011; Valansi et al., 2017; Kumar et al., 2022) may explain why previous studies failed to detect fusion of somatic cells expressing IZUMO1 and JUNO (Bianchi et al., 2014).

### IZUMO1 activity is regulated during fertilization
Despite IZUMO1 being a unilateral fusogen, sperm cells do not normally fuse with other sperm or with other cells in the male or female reproductive tracts. Furthermore, gamete fusion is species specific (Yanagimachi, 1988). Moreover, in fresh sperm, IZUMO1 is not exposed on the surface, being localized to the interior of the acrosome (Inoue et al., 2005), and therefore it cannot mediate cell fusion. Only after capacitation and the acrosomal exocytosis in the female tract, IZUMO1 is exposed to the cell surface and migrates to the fusogenic region of the sperm plasma membrane (Inoue et al., 2005; Satouh et al., 2012). In addition, the requirement of IZUMO-JUNO binding prior to fusion determines that the sperm will fuse to eggs expressing a species-matching JUNO (Bianchi and Wright, 2015). This is a common mechanism used by viruses to regulate their cellular tropism, such as the requirement of CD4 and other co-receptors for HIV infection mediated by the Env glycoprotein (Melikyan, 2011). Another analogy can be made with intracellular fusogens such as the SNARE complex that requires the regulatory activities of Munc18, Munc13, and synaptotagmin for efficient fusion (Stepien and Rizo, 2021). After gamete fusion occurs, JUNO is shed from the plasma membrane preventing further sperm to bind, and therefore, contributing to a block to polyspermy (Bianchi et al., 2014).

### Concluding remarks
Up to this report, GCS1 proteins were the only known fusogens involved in fertilization; these proteins are structurally and evolutionarily related to Class II fusogens from enveloped viruses and FF fusogens from nematodes (Brukman et al., 2022). Together, these fusogens form a superfamily called Fusexins that have recently been identified in archaea, where they may have a primordial function in plasma membrane fusion (Moi et al., 2022). However, many sexually reproducing organisms

including fungi and vertebrates lack a fusexin homologue. Thus, IZUMO1-type proteins are chordates' innovations that may have replaced GCS1 during evolution as a strategy for gamete fusion. We hypothesize that IZUMO1 from the sperm first transiently binds JUNO on the egg for docking and subsequently undergoes a conformational change, oligomerization, and possibly interacts with other proteins (Vondrakova et al., 2022) to induce gamete fusion (Fig. S3 E). This role in membrane merging makes IZUMO1 even more suited to its name, coined after the Japanese shrine dedicated to marriage (Inoue et al., 2005).

## Materials and methods

### Cell lines and DNA transfection
In this study, we used BHK (Cat# CCL-10; ATCC, RRID: CVCL_1915) for multinucleation, content mixing, and BHK-to-egg interaction experiments and for evaluation of expression by Western blot and immunostaining, BHK (Cat# RCB1423; RIKEN Cell Bank, RRID: CVCL_1915) for live imaging assays, and HEK293T (Cat# CRL-3216; ATCC, RRID: CVCL_0063) cells for content mixing experiments. BHK and HEK293T cells were grown and maintained in DMEM containing 10% FBS. Cells were cultured at 37°C in 5% $CO_2$. Plasmids were transfected into cells using 2 μl jetPRIME (PolyPlus-transfection) per μg of DNA in 100 μl of reaction buffer for every ml of medium. For experiments with HEK293T cells, a coating with poly-L-lysine hydrobromide (20 μg/ml; Sigma-Aldrich) was applied to the plates.

### Mice
All animal studies were approved by the Committee on the Ethics of Animal Experiments of the Technion - Israel Institute of Technology. B6D2.C57BL/6-Tg(Zp3EGFP/Cd9)1Osb (Miyado et al., 2008) and B6D2-Tg(Izumo1-mCherry; Satouh et al., 2012) mice lines were obtained from Dr. Masahito Ikawa (Osaka University, Osaka, Japan), and animals were bred and housed in the Technion animal facility under specific pathogen–free conditions with ad libitum access to food and water. The primers used for genotyping are outlined in Table S2. Transgenic EGFP/Cd9 females between 2 and 6 mo and transgenic Izumo1-mCherry male mice between 3 and 6 mo were used for the experiments.

### DNA constructs
For the multinucleation and content mixing assays: Mouse Izumo1 and Juno coding sequences were amplified from pCMV6-IZUMO1-GFP (MG222708; Origene) and pExpress1::JUNO (Clone B2) plasmids, respectively, kindly provided by Gavin Wright. The GFP tag of IZUMO1 construct was replaced with V5-HIS tags during cloning. Izumo1-V5 and Juno sequences and A. thaliana GCS1/HAP2-V5 were subcloned by restriction cloning into pCI::H2B-RFP (Williams et al., 2018) and pCI::GFPnes vectors separately employing enzymes from Thermo Fisher Scientific (Invitrogen). These bicistronic vectors translate for a nuclear RFP (H2B-RFP) or GFPnes after an internal ribosome entry site element. For mutagenesis of IZUMO1: (i) IZUMO1$^{ecto}$: The ectodomain of IZUMO1 (M1-P312) was amplified; (ii) IZUMO1$^{\Delta Ig}$: The

upstream and downstream of Ig domain (G166-L253) were amplified independently from the full-length IZUMO1 and fused together by overlap PCR; (iii) IZUMO1$^{W148A}$: The W148 was mutated to alanine by overlapping PCR; (iv) IZUMO1$^{FWW}$: F28, W88, and W113 located in the Izumo domain were mutated to alanines by overlapping primers and fused together. The folding of the mutants was corroborated by AlphaFold (Jumper et al., 2021). All mutants were ligated into pCI::H2B-RFP and pCI::GFPnes vectors for mixing assay. JUNO was tagged in the C terminus by inserting an annealed oligo containing the FLAG sequence after the signal peptide into the BlpI restriction site. Oligonucleotides were obtained from Sigma-Aldrich or IDT and all constructs were verified by DNA sequencing (Macrogen). For the live imaging experiments, *A. thaliana GEX2*, *C. elegans eff-1*, and mouse *Izumo* or *Juno* sequences were amplified from cDNAs. To visualize the proteins, the fragments corresponding to *GEX2* and *Izumo1* were fused to fluorescent protein sequences by PCR and then cloned into pGENE B after double digestion with restriction enzymes using the Gibson assembly (#E5510; NEB). pGENE::mCherry-JUNO was made by ligating mCherry and pGENE::JUNO fragments (Nakajima et al., 2022). For inducible expression using mifepristone in BHK cells, we used the Gene-Switch System (Invitrogen). For lentiviral transduction, the sequence for IZUMO1-GFP was cloned into the pLVX-TetOne-Puro vector (#631849; Clontech). The complete lists of primers and plasmids used in this study are shown in Tables S2 and S3, respectively.

### Immunostaining and analysis of localization
BHK cells were grown on 24-well glass bottom tissue-culture plates or in tissue-culture plates with coverslips. 24 h after plating, cells were transfected and incubated for additional 24 h before proceeding to the immunostaining. For IZUMO1 (WT) and its mutants (IZUMO1$^{ecto}$, IZUMO1$^{\Delta Ig}$, IZUMO1$^{W148A}$, and IZUMO1$^{FWW}$) encoded in the pCI::H2B-RFP vector cells were fixed with 4% PFA in PBS and, when indicated, permeabilized with 0.1% Triton X-100 in PBS. To detect the proteins, immunofluorescence was performed with anti-V5 (1:500, Cat# R96025; Thermo Fisher Scientific, RRID: AB_2556564) or anti-IZUMO1, clone Mab120 (1:500, Cat# MABT1357; Merck Millipore) followed by the secondary antibodies Alexa Fluor 488 goat anti-mouse (1:500, Cat# A-21202; Thermo Fisher Scientific, RRID: AB_141607) and Alexa Fluor 488 chicken anti-rat (1:500, Cat# A-21470; Thermo Fisher Scientific, RRID: AB_2535873), respectively. For JUNO localization, pExpress1::JUNO-flag plasmid was transfected together with the empty pCI::H2B-RFP and the incubation with the first antibody anti-flag (1:1,000, Cat# F3165; Sigma-Aldrich, RRID: AB_259529) was performed before fixation (for detecting surface protein) or after fixation and permeabilization (for detecting total protein). Then, JUNO was probed using the secondary Alexa Fluor 488 goat anti-mouse. In all cases, the nuclei were stained with 1 µg/ml DAPI and micrographs were obtained using wide-field illumination using an ELYRA system S.1 microscope (Plan-Apochromat 20× NA 0.8; Zeiss) with an EMCCD iXon camera (Andor) through ZEN microscopy software 7.0.4.0 (RRID: SCR_013672; Zeiss).

### Evaluation of multinucleation
BHK cells were grown on 24-well glass bottom tissue-culture plates. 24 h after plating, cells were transfected with pCI::myrGFP::H2B-RFP (myrGFP; Dunsing et al., 2018), pCI::GCS1/HAP2-V5::H2B-RFP, pCI::IZUMO1-V5::H2B-RFP, or pCI::JUNO::H2B-RFP vectors encoding for myrGFP (gray), GCS1/HAP2-V5, IZUMO1-V5, or JUNO. For JUNO, a plasmid for myrGFP (gray) was co-transfected. 24 h after transfection; 20 µM FdUrd was added to the plates to arrest the cell cycle and 24 h later, the cells were fixed with 4% PFA in PBS and processed for immunofluorescence using an anti-V5 antibody, as explained before. Micrographs were obtained using wide-field illumination using an ELYRA system S.1 microscope (Plan-Apochromat 20× NA 0.8; Zeiss). Multinucleation percentage was determined as the ratio between the number of nuclei in multinucleated cells (NuM) and the total number of nuclei in multinucleated cells and expressing cells that were in contact but did not fuse (NuC) as follows: (NuM/[NuC + NuM]) × 100. In some cases, confocal images were acquired using a CSU-W1 spinning disk (Yokogawa) on a Nikon Ti-2 inverted microscope (water Plan-Apochromat 60× NA 1.2; Nikon), equipped with a Prime BSI sCMOS camera (Teledyne Photometrics) through NIS-Elements AR software (RRID: SCR_014329; Nikon).

In another setting, a BHK cell line with inducible expression of IZUMO1-GFP was generated by lentiviral transduction of pLVX-TetOn-IZUMO1-GFP plasmid. Lentiviral particles were produced by transfection of HEK293T cells with 5 µg of pLVX-TetOn-IZUMO1-GFP together with 3.75 µg of the packaging plasmid psPAX2 (#12260; Addgene) and 1.25 µg of the VSV-G envelope expressing plasmid pMD2.G (#12259; Addgene). 48 h later, the supernatant was collected, filtered through 0.45-µm syringe filters, and used for transduction of BHK cells. The transduced cells were selected with 4 µg/ml puromycin for 72 h. For the evaluation of multinucleation, WT or stable lines induced with different concentrations of doxycycline (0, 0.25, 0.5, and 1 µg/ml) were incubated for 48 h. The doxycycline-induced cells were analyzed by direct detection of IZUMO1-GFP; WT and uninduced cells were stained with CellMask Green (#C37608; Invitrogen) for 15 min before fixation. Multinucleation was evaluated as explained above.

### Content mixing experiments
BHK or HEK293T cells at 70% confluence in 35-mm plates were transfected with 1 µg pCI::H2B-RFP or pCI::GFPnes (empty vectors as negative controls); pCI::GCS1/HAP2-V5::H2B-RFP or pCI::GCS1/HAP2-V5::GFPnes (positive control); pCI::IZUMO1-V5::H2B-RFP or pCI::IZUMO1-V5::GFPnes; pCI::JUNO::H2B-RFP or pCI::JUNO::GFPnes. 4 h after transfection, the cells were washed four times with DMEM with 10% serum, four times with PBS, and detached using Trypsin (Biological Industries). The transfected cells were collected, resuspended in DMEM with 10% serum, and counted. Equal amounts of H2B-RFP and GFPnes cells (1–1.25 × 10$^5$ each) were mixed and seeded on glass-bottom plates (12-well black, glass-bottom #1.5H; Cellvis) and incubated at 37°C and 5% CO$_2$. For IZUMO1, pCI::IZUMO-V5::GFPnes cells were also mixed with pCI::H2B-RFP or pCI::JUNO::H2B-RFP transfected cells. 18 h after mixing, 20 µM FdUrd was added to

the BHK cells. The mixed cells were co-incubated for a total of 48 h after which they were fixed with 4% PFA in PBS and stained with 1 µg/ml DAPI. Micrographs were obtained using wide-field illumination using an ELYRA system S.1 microscope (Plan-Apochromat 20× NA 0.8; Zeiss). The percentage of mixing was defined as the ratio between the nuclei in mixed cells (NuM) and the total number of nuclei in mixed cells and fluorescent cells in contact that did not fuse (NuC), as follows: % of mixing = (NuM/[NuM + NuC]) × 100. 1,000 nuclei (NuM + NuC) were counted in each independent repetition (experimental point). For immunostaining after the content mixing, cells were treated as explained above, using as the secondary antibody, the Alexa Fluor 647 goat anti-mouse (Cat# A21235; Thermo Fisher Scientific, RRID: AB_2535804).

## Live imaging experiments
To evaluate fusion by live imaging, we transfected BHK cells with pGENE and pSWITCH. 24 h after transfection, BHK cells were cultured at $5.0 × 10^4$ cells/ml. 4 h after transfection, the expression was induced by addition of $10^{-4}$ mM mifepristone. 3–4 h after induction, images of the cells were acquired every 6 min for 12 h to record cell-to-cell fusion, using a spinning disk confocal system (CellVoyager CV1000; Yokogawa Electric) at a magnification of 10× (NA 0.40, 10×UPLSAPO; Olympus) dry objective. Five different focal planes were registered for each time point. The number of transfected cells and the occurrence of fusion were evaluated. Image analyses were performed using CV1000 software (Yokogawa Electric), and the FIJI online tool was used to adjust the brightness and contrast. The percentage of fusion was defined as the ratio between the number of fusion events (Fe) and the number of transfected cells (Tc), as follows: % of fusion = Fe/Tc.

## Surface biotinylation
BHK cells at 70% confluence in 35-mm plates were transfected with 1 µg pCAGGS::IZUMO-V5 (naive cells as negative control); pCAGGS::IZUMO1$^{ecto}$-V5, pCAGGS::IZUMO1$^{ΔIg}$-V5, pCAGGS::IZUMO1$^{W148A}$-V5, or pCAGGS::IZUMO1$^{FWW}$-V5. 24 h later, cells were washed twice with ice-cold PBS$^{2+}$ (with Ca$^{2+}$ and Mg$^{2+}$) and incubated with 0.5 mg/ml EZ-Link Sulfo NHS-Biotin (#21217; Thermo Fisher Scientific) for 30 min on ice. The cells were washed four times with ice-cold PBS$^{2+}$, once with DMEM with 10% FBS (to quench residual biotin), followed by two more washes with PBS$^{2+}$. The cells were resuspended in 300 µl of Lysis Buffer supplemented with 10 mM iodoacetamide and detached using a scrapper. The insoluble debris was separated by centrifugation (10 min at 21,000 $g$); for total protein, 15 µl of the supernatant was mixed with reducing sample buffer (#S3401; Sigma-Aldrich) and incubated 5 min at 95°C. For surface biotinylated protein, the lysate was mixed with NeutrAvidin Agarose Resin (#29202; Thermo Fisher Scientific) and 0.3% SDS. After an incubation of 12 h at 4°C, the resin was centrifuged (2 min at 21,000 $g$), washed three times with lysis buffer, and then mixed with SDS-PAGE loading solution with freshly added 5% β-mercaptoethanol and incubated 5 min at 95°C. After pelleting by centrifugation, samples were loaded on a BOLT 4–12% Bis-Tris Plus gel and transferred to polyvinylidene fluoride

membrane for 1 h, membranes blocked (EZ block, #41-805-10; Biological Industries), and incubated with primary antibody anti-V5 mouse monoclonal antibody or anti-actin (1:2,000, Cat# ICN691001; MP Biomedicals, RRID: AB_2335127) for 1 h at room temperature and HRP-conjugated goat anti-mouse secondary antibody (1:15,000, Cat# 115-035-003; Jackson ImmunoResearch Labs, RRID: AB_10015289) for 1 h at room temperature. Membranes were imaged by the ECL detection system (#34078; Thermo Fisher Scientific) using FUSION-PULSE.6 (VILBER).

## Fusion of BHK cells to sperm
BHK cells at 70% confluence in 35-mm plates were transfected with 0.5 µg pcDNA3.1-EGFP-MBD-nls alone or together with 1 µg pExpress1::JUNO 24 h before the collection of the sperm. The plasmid pcDNA3.1-EGFP-MBD-nls for an EGFP fused to the methylcytosine binding domain and nuclear localization signal of human MBD1 (Yamagata et al., 2005). Sperm were recovered from the epididymides of *Izumo1-mCherry* transgenic mice into 300 µl of mHTF medium (Kito et al., 2004) supplemented with 4 mg/ml of BSA. The sperm were diluted in fresh medium to a concentration of $25 × 10^6$ cells/ml and incubated for 90–120 min at 37°C and 5% CO$_2$ to induce capacitation. The medium of the BHK cells was replaced with the sperm suspension in mHTF ($2 × 10^6$ cells/ml) and incubated for 4 h at 37°C and 5% CO$_2$. Later, the cells were washed with fresh medium, fixed with 4% PFA in PBS and stained with 1 µg/ml DAPI. Micrographs were obtained using wide-field illumination using an ELYRA system S.1 microscope (Plan-Apochromat 63× NA 1.4; Zeiss). Fusion between BHK and sperm cells was evaluated by the presence of green fluorescent sperm heads due to the transfer of EGFP-MBD-nls from the BHK cytoplasm (Fig. S3 A), as it was previously described for sperm–egg fusion (Mori et al., 2021).

## Fusion of BHK cells to oocytes
BHK cells at 70% confluence in 35-mm plates were transfected with 1 µg pCI::IZUMO1::H2B-RFP alone or together with pMD2.G (encoding for VSV-G) 24 h before the collection of the oocytes. To induce ovulation, transgenic females were treated with an i.p. injection of pregnant mare serum gonadotropin (5 IU; #HOR-272, Prospec), followed by an i.p. injection of human chorionic gonadotropin (5 IU, #CG5; Sigma-Aldrich) 48 h later. Cumulus–oocyte complexes were collected from the ampullae of induced females 12–15 h after human chorionic gonadotropin administration in mHTF medium. The oocytes were denuded from the cumulus and the ZP by sequential treatment with 0.3 mg/ml hyaluronidase (H3506; Sigma-Aldrich) and acid Tyrode solution (pH 2.5; Nicolson et al., 1975). BHK cells were harvested with 0.05% EDTA in PBS and washed once with mHTF. 15–20 ZP-free oocytes were incubated with $5 × 10^4$ BHK cells for 15 min with occasional mixing and washed. In the case of VSV-G–expressing cells, the oocytes were treated for 30 s with acid Tyrode solution (pH 2.5) and washed again with mHTF. In all cases, the oocytes–cells complexes were incubated for 5 h at 37°C and 5% CO$_2$. Oocytes were then fixed, stained with DAPI and evaluated for the presence of H2B-RFP chromosomes within the cytoplasm (Fig. S3 C), using a CSUX1 spinning disk confocal (Yokogawa) on a Nikon Eclipse Ti inverted microscope

(Plan-Apochromat 60× NA 1.4, Nikon). Images were obtained using an iXon3 EMCCD camera (ANDOR) through MetaMorph (Molecular Devices, version 7.8.1.0).

### Binding of BHK cells to oocytes

BHK cells at 70% confluence in 35-mm plates were transfected with 1 µg pCI::GFPnes, pCI::IZUMO1::GFPnes, pCI::IZUMO1$^{W148A}$::GFPnes, or pCI::IZUMO1$^{FWW}$::GFPnes 24 h before the collection of the oocytes. Oocytes were obtained from WT females as described above, incubated with the $5 \times 10^4$ BHK cells with occasional mixing for 15 min, washed and fixed. The average number of BHK in direct contact with the oocyte was determined for each group. Micrographs were obtained using wide-field illumination as described above.

### Statistics and data analysis

Results are presented as means ± SEM. For each experiment, we performed at least three independent biological repetitions. To evaluate the significance of differences between the averages, we used one-way ANOVA as described in the legends (GraphPad Prism 9, RRID: SCR_002798). Figures were prepared with Photoshop CS6 (Adobe, RRID: SCR_014199), Illustrator CS6 (Adobe, RRID: SCR_010279), BioRender.com (RRID: SCR_018361), and FIJI (ImageJ 1.53c, RRID: SCR_002285).

### Online supplemental material

Fig. S1 shows surface localization of IZUMO1 and JUNO; induction of syncytia formation by IZUMO1, related to Fig. 1. Fig. S2 shows content mixing assays of BHK and HEK293T cells, related to Fig. 2. Fig. S3 shows sperm fusion to BHK-JUNO cells and oocyte fusion to BHK-IZUMO1+VSV-G cells; summary and working model, related to Figs. 2, 3, 4, and 5. Table S1 shows syncytia formation, related to Fig. 1. Table S2 shows primers used, related to Materials and methods. Table S3 shows plasmids used, related to Materials and methods. Video 1 shows Z-Series of tetranucleate IZUMO1-V5 cell from Fig. 1 D. Video 2 is a time-lapse animation of EFF-1–expressing cells undergoing bilateral fusion. Video 3 shows IZUMO1-expressing cells undergoing bilateral fusion, corresponding to Fig. 3 A. Video 4 shows the single green channel of Video 3. Video 5 shows bright field channel of Video 3. Video 6 shows different bright field focal planes at 7:48 of Video 5. Video 7 shows the crystal structure of IZUMO1 (Nishimura et al., 2016, PDB: 5B5K) related to Fig. 4 B showing the solvent-excluded surface of the ectodomain of IZUMO1.

## Acknowledgments

We thank Gavin Wright (University of York, York, UK) for *mIzumo1* and *mJuno* plasmids, Kazuo Yamagata (Kindai University, Higashiosaka City, Osaka, Japan) for the EGFP-MBD-NLS plasmid, and Masahito Ikawa (Osaka University, Osaka, Japan) for the CD9-GFP and IZUMO1-mCherry transgenic mouse lines. We thank Pablo Aguilar, Dan Cassel, Masahito Ikawa, Yael Iosilevskii, and Luca Jovine for critically reading the manuscript.

This work was funded by the Israel Science Foundation (257/17, 2462/18, 2327/19, and 178/20 to B. Podbilewicz), by Japan Society for the Promotion of Science Grant-in-Aid for Scientific Research on Innovative Areas (16H06465, 16H06464, and 16K21727 to T. Higashiyama), by Japan Science and Technology Agency Exploratory Research for Advanced Technology (JPMJER1004 to T. Higashiyama) and Core Research for Evolutionary Science and Technology (JPMJCR20E5 to T. Higashiyama). This project has received funding from the European Union's Horizon 2020 research and innovation program under the Marie Skłodowska-Curie (844807 to N.G. Brukman).

The authors declare no competing financial interests.

Author contributions: B. Podbilewicz and T. Higashiyama conceived the study. N.G. Brukman, K.P. Nakajima, C. Valansi, X. Li, and K. Flyak designed and conducted experiments and processed and analyzed the data supervised by B. Podbilewicz and T. Higashiyama. N.G. Brukman and B. Podbilewicz wrote the initial draft of the manuscript. All authors participated in discussions of results and manuscript editing.

Submitted: 29 July 2022

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

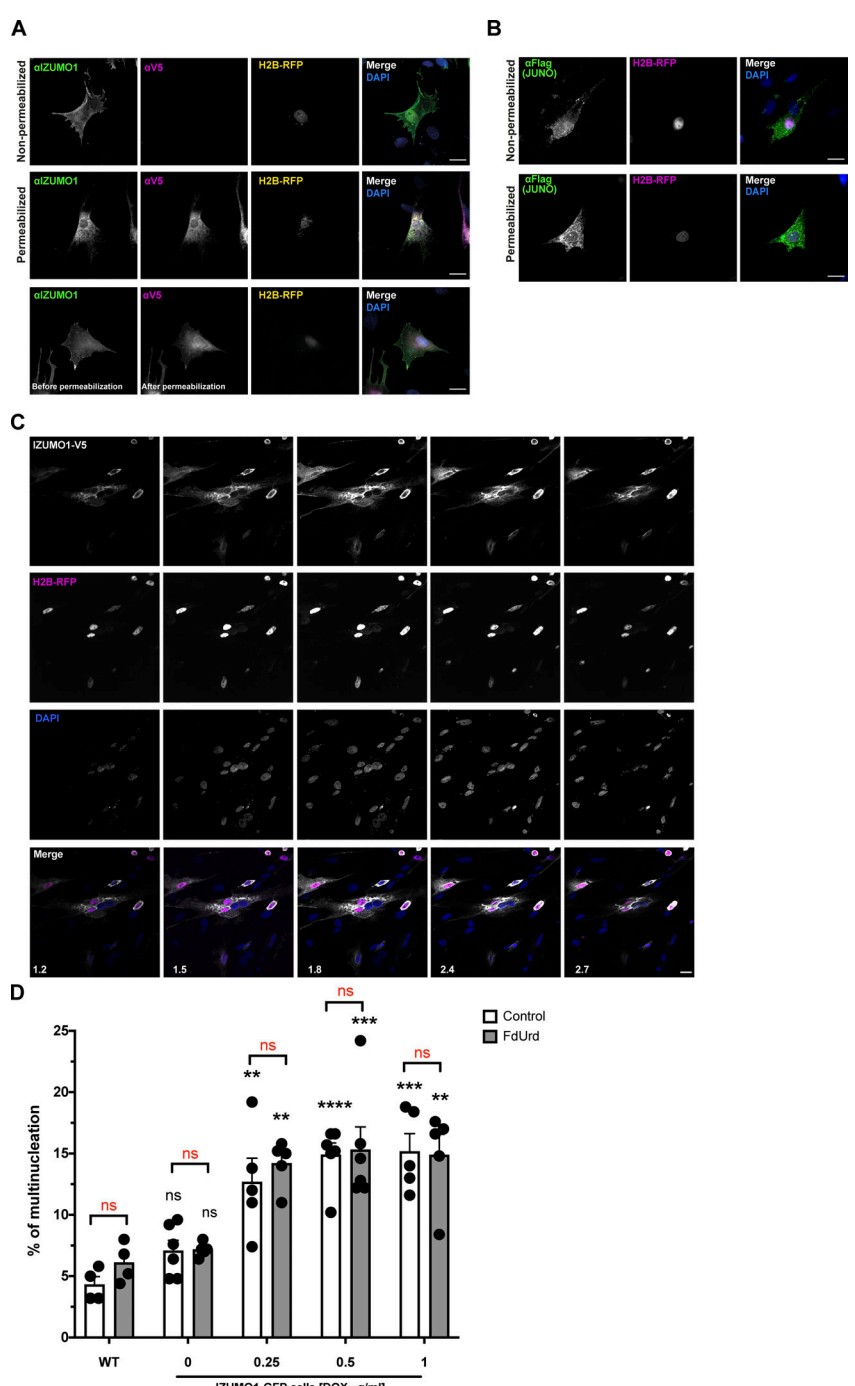

**Figure S1.** **Surface localization of IZUMO1 and JUNO; induction of multinucleation of BHK cells by IZUMO1, related to** Fig. 1**. (A)** IZUMO1 presence on the surface of BHK cells transfected with pCI::IZUMO1-V5::H2B-RFP was determined by immunostaining using an anti-IZUMO1 (Mab 120) that recognizes the extracellular N-terminal region of the protein (green) or anti-V5 against the cytoplasmic C-terminal tag (magenta). Non-permeabilized and permeabilized cells were tested. Non-permeabilized cells were exposed to anti-IZUMO1, then permeabilized and finally incubated with anti-V5 (lower row). In all cases, DAPI staining is shown in blue in the merge. Scale bars, 20 μm. **(B)** JUNO localization to the plasma membrane of BHK transfected with pExpress1::JUNO-flag and pCI::H2B-RFP was evaluated by incubating the cells with anti-flag antibody before fixation (non-permeabilized) or after fixation and permeabilization. The immunostaining signal (green) and the nuclei of transfected cells (magenta) are shown in separate channels and in the merge, which includes the DAPI staining (blue). Scale bars, 20 μm. **(C)** Images from Fig. 1 D in each separate channel (green in gray, magenta, and DAPI) and merge. The numbers in the merged images correspond to the confocal planes in micrometers. Note that two nuclei in magenta and two in blue suggest the recent fusion between cells expressing IZUMO1::H2B-RFP (magenta) with untransfected cells (blue). See also Video 1. Scale bar, 20 μm. **(D)** WT or cells with doxycycline (DOX)-inducible expression of IZUMO1-GFP were evaluated for multinucleation in the presence or absence of the inhibitor FdUrd. Different concentrations of DOX were tested. WT and uninduced cells were stained with CellMask Green. The percentage of multinucleation was defined as the ratio between the nuclei in multinucleated cells (NuM) and the total number of nuclei in multinucleated and fluorescent cells in contact that did not fuse (NuC), as follows: % of multinucleation = (NuM/[NuM + NuC]) × 100. We show individual data and means ± SEM of four to six independent experiments. Comparisons were made with two-way ANOVA followed by Bonferroni's multiple comparisons against the WT (in black) or between FdUrd and control (in red). **P < 0.01, ***P < 0.001, ****P < 0.0001.

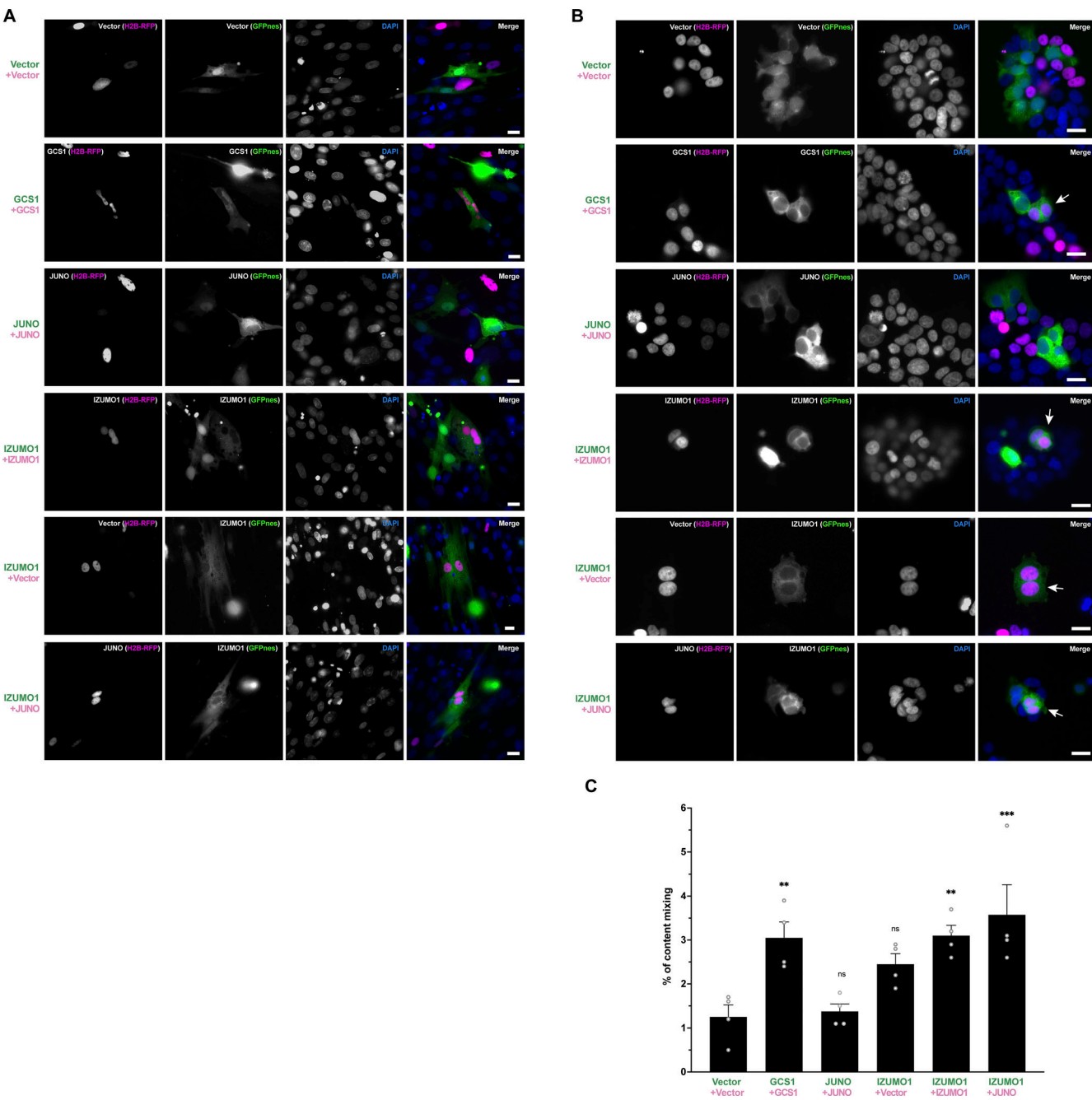

**Figure S2. Content mixing assays of BHK and HEK293T cells. (A)** Evaluation of content mixing of BHK cells. Images from Fig. 2 A in each separate channel (red, green, and DAPI) and merge. Scale bars, 20 µm. **(B)** Evaluation of cell fusion by content mixing of human embryonic kidney HEK293T cells, related to Fig. 2. Images of mixed cells transfected with pCI::GFPnes or pCI::H2B-RFP empty vectors or containing the coding sequence for the expression of GCS1, IZUMO1, and JUNO as indicated in each panel. Arrows show fused cells containing green cytoplasm (GFPnes) with red nuclei (H2B-RFP). DAPI staining, blue. Each channel (red, green, and DAPI) and merge are shown. Scale bars, 20 µm. **(C)** Quantification of content mixing of HEK293T cells. The percentage of mixing is the ratio between the nuclei in mixed cells (NuM) and the total number of nuclei in mixed cells and fluorescent cells in contact that did not fuse (NuC), as follows: % of content mixing = (NuM/[NuM + NuC]) × 100. Graph shows experiment values and means ± SEM of four independent experiments. Comparisons by one-way ANOVA followed by Dunnett's test against the empty vectors. **P < 0.01, ***P < 0.001.

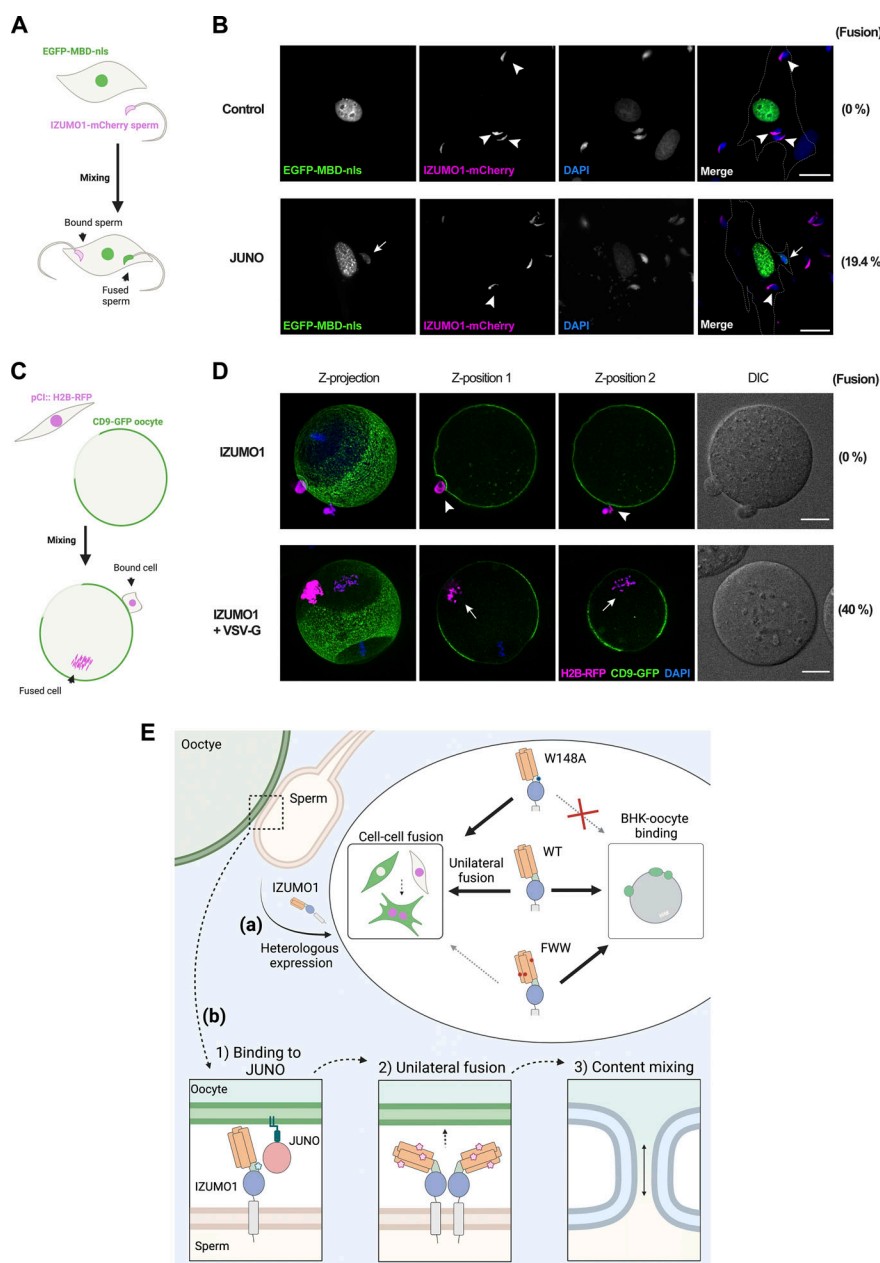

Figure S3.   **Fusion between BHK-JUNO to sperm and BHK-IZUMO1+VSV-G to oocyte fusion. (A and B)** BHK cells transfected with pcDNA3.1-EGFP-MBD-nls alone (control) or together with pExpress1::JUNO were incubated with IZUMO1-mCherry sperm (magenta) and observed by fluorescence microscopy. Fusion was evaluated as the presence of EGFP-MBD-nls signal (green) inside the sperm heads (arrows) as opposed to bound cells (arrowheads). 175 out of 900 cells expressing JUNO presented at least one sperm fused to them (*n* = 3), while none of 900 control cells showed signs of fusion (*n* = 3). None of EGFP-MBD-NLS–positive (fused) sperm showed IZUMO1-mCherry signal, while 85% (*n* = 300) of EGFP-MBD-nls–negative (but bound) sperm carried IZUMO1-mCherry. The outline of the cells is represented with a dashed line. DAPI staining (blue). Scale bars, 20 μm. **(C and D)** BHK cells transfected with pCI::IZUMO1::H2B-RFP alone or together with a plasmid encoding for VSV-G were incubated with CD9-GFP oocytes (green) and observed by confocal microscopy. Fusion was evaluated as the presence of RFP-positive chromosomes (magenta) inside the cytoplasm of the eggs (arrows) as opposed to bound cells (arrowheads). For IZUMO1, no fusion was detected in >100 oocytes with cells attached, while for IZUMO1+VSV-G, 18 out of 45 oocytes with cells bound displayed fusion. To activate VSV-G activity, we exposed these oocytes to pH 2.5 for 30 s. The maximal intensity Z-projection, two different focal planes (Z-position 1 and 2), and DIC are shown. DAPI staining (blue). Scale bars, 20 μm. **(E)** Summary and working model. **(a)** Heterologous IZUMO1 expression in somatic cells in culture induces cell-to-cell fusion in a unilateral manner and independently of JUNO. The mutation W148A had no effect on the levels of content mixing compared to WT IZUMO1, but disrupted the binding of BHK cells to oocytes. The mutant FWW was still able to mediate BHK binding to the oocyte; however, it induced significantly lower levels of content mixing than WT IZUMO1. These results support the existence of two functional domains in IZUMO1, with binding to JUNO mediated through W148 on the hinge and cell-to-cell fusion dependent on three exposed aromatic residues on the four-helix bundle of the Izumo domain. **(b)** Working model for IZUMO1 activities. 1. Transient interaction between JUNO and IZUMO1 ("Binding") mediated by IZUMO1 W148 on the hinge region (blue star). 2. Conformational change of IZUMO1, oligomerization (Inoue et al., 2015), and unilateral induction of gamete fusion ("Unilateral fusion"), enhanced by the action of F28, W88, and W113, the surface exposed aromatic residues on the four-helix bundle of IZUMO1 (pink stars). The action of secondary receptor(s) for IZUMO1 is also possible. IZUMO1 domain colors correspond to Fig. 4, A and C. 3. Following pore formation, there is cytoplasmic mixing.

Video 1.   **Z-Series of tetranucleate IZUMO1-V5 cell from** Fig. 1 D**.** Each confocal section is 0.3 µm apart. The video is shown at 3 frames/s.

Video 2.   **Time-lapse animation of EFF-1–expressing cells (magenta) shows bilateral fusion, corresponding to** Fig. 3 A**.** Time (h:min). Scale bar, 50 µm. The video is shown at 7 frames/s.

Video 3.   **IZUMO1-expressing cells (green) show bilateral fusion, corresponding to** Fig. 3 A**.** Note metaphase stage of mitosis at 3:24 and sister cells completely separated at 7:48. At 8:30 both cells express IZUMO1-Venus and cytoplasmic mixing can be seen at 9:00, about 6 h after the initiation of mitosis. Time (h:min). Scale bar, 50 µm. The video is shown at 7 frames/s.

Video 4.   **Single green channel of** Video 3**.** Time (h:min). Scale bar, 50 µm. The video is shown at 7 frames/s.

Video 5.   **Bright field channel of** Video 3**.** Time (h:min). Scale bar, 50 µm. The video is shown at 7 frames/s.

Video 6.   **Different bright field focal planes at 7:48 of** Video 5 **showing that the two mononucleated cells (arrowheads) appear to be unlinked before fusion.** Time (h:min). Scale bar, 50 µm. The video is shown at 7 frames/s.

Video 7.   **Crystal structure of IZUMO1 (**Nishimura et al., 2016**, PDB:** 5B5K**), related to** Fig. 4 B**.** Solvent-excluded surface of the ectodomain is colored based on molecular lipophilicity potential maps, ranging from dark cyan (hydrophilic) to dark gold (lipophilic). The aromatic residues F28, W88, and W113 are contoured in black. Video displays at 25 frames/s.

**Provided online are three tables. Table S1 shows GCS1/HAP2 and IZUMO1 induce syncytia formation. Table S2 lists primers used for this study. Table S3 lists plasmids used for this study.**

