## [Peer Review File · The Journal of Cell Biology]

A novel function for the sperm adhesion protein IZUMO1 in cell-cell fusion

Nicolas Brukman, Kohdai Nakajima, Clari Valansi, Kateryna Flyak, Xiaohui Li, Tetsuya Higashiyama, and Benjamin Podbilewicz

Corresponding Author(s): Benjamin Podbilewicz, Technion - Israel Institute of Technology and Tetsuya Higashiyama, Department of Biological Sciences, the University of Tokyo

Review Timeline:

Submission Date:	2022-07-29
Editorial Decision:	2022-09-07
Revision Received:	2022-10-11
Editorial Decision:	2022-11-02
Revision Received:	2022-11-04

Monitoring Editor: Michael Rudnicki

Scientific Editor: Tim Fessenden

Transaction Report:

DOI: <https://doi.org/10.1083/jcb.202207147>

September 7, 2022

Re: JCB manuscript #202207147

Prof. Benjamin Podbilewicz
Technion - Israel Institute of Technology
Biology
Technion City
Haifa 32000
Israel

Dear Prof. Podbilewicz,

Thank you for submitting your manuscript entitled "A novel function for the sperm adhesion molecule IZUMO1 as a cell-cell fusion protein". The manuscript was assessed by expert reviewers, whose comments are appended to this letter. We invite you to submit a revision if you can address the reviewers' key concerns, as outlined here.

As you will see, all reviewers found this work provided an important advance by defining Izumo1 as a mammalian fusogen and all support publication. Requests made by reviewers focused on clarifying assays in place. One reviewer was concerned that the role of this protein is left unclear by the heterologous cell fusion assay using oocytes. While not a requirement for publication, we encourage the inclusion of additional data if possible to better describe the role of Izumo1 in sperm-egg fusion. Please note that we will send the revised manuscript to be assessed by at least two reviewers.

GENERAL GUIDELINES:

Text limits: Character count for a Report is < 20,000, not including spaces. Count includes title page, abstract, introduction, the joint Results & Discussion, and acknowledgments. Count does not include materials and methods, figure legends, references, tables, or supplemental legends.

Figures: Reports may have up to 5 main text figures. To avoid delays in production, figures must be prepared according to the policies outlined in our Instructions to Authors, under Data Presentation, <https://jcb.rupress.org/site/misc/ifora.xhtml>. All figures in accepted manuscripts will be screened prior to publication.

Supplemental information: There are strict limits on the allowable amount of supplemental data. Reports may have up to 3 supplemental figures. Up to 10 supplemental videos or flash animations are allowed. A summary of all supplemental material should appear at the end of the Materials and methods section.

Please note that JCB now requires authors to submit Source Data used to generate figures containing gels and Western blots with all revised manuscripts. This Source Data consists of fully uncropped and unprocessed images for each gel/blot displayed in the main and supplemental figures. Since your paper includes cropped gel and/or blot images, please be sure to provide one Source Data file for each figure that contains gels and/or blots along with your revised manuscript files. File names for Source Data figures should be alphanumeric without any spaces or special characters (i.e., SourceDataF#, where F# refers to the associated main figure number or SourceDataFS# for those associated with Supplementary figures). The lanes of the gels/blots should be labeled as they are in the associated figure, the place where cropping was applied should be marked (with a box), and molecular weight/size standards should be labeled wherever possible. Source Data files will be made available to reviewers during evaluation of revised manuscripts and, if your paper is eventually published in JCB, the files will be directly linked to specific figures in the published article.

The typical timeframe for revisions is three to four months. While most universities and institutes have reopened labs and allowed researchers to begin working at nearly pre-pandemic levels, we at JCB realize that the lingering effects of the COVID-19 pandemic may still be impacting some aspects of your work, including the acquisition of equipment and reagents. Therefore,

if you anticipate any difficulties in meeting this aforementioned revision time limit, please contact us and we can work with you to find an appropriate time frame for resubmission. Please note that papers are generally considered through only one revision cycle, so any revised manuscript will likely be either accepted or rejected.

Thank you for this interesting contribution to Journal of Cell Biology. You can contact us at the journal office with any questions, cellbio@rockefeller.edu or call (212) 327-8588.

Sincerely,

Michael Rudnicki
Monitoring Editor
Journal of Cell Biology

Tim Fessenden
Scientific Editor
Journal of Cell Biology

Reviewer #1 (Comments to the Authors (Required)):

Considerable work has implicated the sperm protein Izumo and its egg receptor, Juno, in mammalian sperm-egg fusion. Work has also documented localization dynamics and structural transitions in Izumo that precede fusion (Inoue, 2015; see model in Fig. 6, Inoue, 2015). However, prior studies have shown that while somatic cells expressing Izumo bound to somatic cells expressing Juno, they did not fuse (Bianchi 2014; Inoue, 2015); similarly somatic cells expressing Izumo bound to, but did not fuse with, mammalian eggs (Inoue 2015). Even cells expressing Izumo plus four other sperm factors, or Izumo plus DCST1 and DSCT2, failed to fuse with oocytes (see Noda 2022).

The current study demonstrates fusion activity for Izumo for the first time. BHK cells expressing Izumo fuse to BHK cells expressing Juno (and to mock transfected BHK cells!), as shown in both syncytia (Fig. 1), and content mixing (Figs. 2 and 3) assays. The authors also provide mutational evidence for a role for three exposed aromatic residues for BHK-Izumo-to-BHK-Juno fusion (Fig. 5: content mixing assay). But, as shown previously (e.g., Inoue 2015), BHK-Izumo cells bind to, but do not fuse with, oocytes (Fig. S4). So, this study demonstrates the fusion activity of Izumo for the first time, but how this happens mechanistically in the context of sperm and eggs remains to be elucidated.

Main Comment

While mutation of 3 aromatic residues (F28, W88, and W113) diminishes the fusion activity of Izumo, these residues do not appear (from Fig. 4A) to form any type of fusion peptide, fusion loop, or fusion (hydrophobic) patch* (as stated on Lines 276-277 in the Discussion) that might mediate protein-lipid interactions. (a) Perhaps a space-filling model colored on a hydrophobicity scale would be informative as part of Fig. 4. (b) Also, please make this point in the Results section when discussing the triple mutant. (c) It is suggested to edit the sentence on Lines 375-377 of the Discussion. The author's model (Fig. S5) is quite in line with the model proposed in Fig. 6 of Inoue 2015, albeit without proposing a second oocyte receptor for Izumo. (d) The work would be stronger if there were insight into what the three aromatic residues do to promote fusion: e.g., testing some of the proposals raised on Lines 291-295 of the Discussion. [*All fusogens (viral and cellular), as well as the reovirus FAST proteins, contains a 'coherent' hydrophobic region (fusion peptide or fusion loop) that has (or has been strongly implicated) to mediate the fusogen-lipid bilayer interaction that precedes fusion.]

Minor Comments

1. Line 158: This could be stated more strongly. All tested viral fusion proteins (and the related syncytins) can induce fusion when expressed in only one cell (donor cell) of a set of cells. This is also true for the reovirus FAST proteins.
2. Reovirus FAST proteins seem to warrant a mention in this manuscript, as they are structurally distinct from fusogens and Izumo, but are still sufficient to mediate fusion (i.e., are fusogens).
3. Fig. 3A legend. Clarify/add arrow(head)s. The reader does not see 'two individual fusion events'.
4. Line 194: Suggested edit: 'unilateral mechanism to fuse somatic mammalian cells'.
5. Figs. S2 and S3 appear to be the same. The legends say that S2 is with BHK cells and S3 is with 293T cells.
6. Results section (especially re: Fig. 5): Please further specify which binding and fusion assays were employed for each

statement (e.g., BHK-oocyte binding; BHK-BHK content mixing fusion; BHK-oocyte fusion).

7. Line 241: You might add: 'as is the case for viral fusion proteins' (and FFs if tested).

8. Fig. S4: It is suggested to provide quantitation in the figure proper (currently now in legend). You could put % fused (of bound) cells in the picture boxes or provide a new panel with the quantitation.

9. Fig. S4 legend: Mention that cell complexes were exposed to low (?what) pH to activate VSV G? Were samples in all panels of Fig. S4 exposed to low pH (e.g., ones in top panels, Izumo alone)? This wasn't clear. Relatedly, does low pH treatment inactivate the fusion activity of Izumo? If this hasn't been assessed, can you test that with one of your BHK Izumo fusion assays? While not deemed highly likely, it would be good to rule out this possibility.

10. Line 218: Provide a reference (and PDB #) for the Izumo crystal structure.

11. Line 227: Suggested edit: 'like certain members of the Fusexin family'. (Not all viral fusion proteins have IgG domains.)

12. Line 273: Specify 'heterologous somatic cell-to-cell fusion' (or 'BHK cell-to-cell fusion'), since BHK-Izumo cells did not fuse with oocytes.

13. Line 279: Clarify/elaborate: Which fusexins do not have a fusion peptide or fusion loop? How do they function in fusion without such an element?

Reviewer #2 (Comments to the Authors (Required)):

Sperm protein IZUMO1 is essential for sperm-egg binding in mammalian fertilization. This binding is followed by cell fusion and the identity of the protein(s) that fuse the plasma membranes of sperm and egg has remained one of the most important open questions in the field of fertilization. Brukman et al. show that expression of IZUMO1 in mammalian cells that do not normally fuse is sufficient to induce their fusion detected as formation of multinucleated cells and content mixing. This work corrects an earlier belief in the field that IZUMO1 does not fuse cells based on just one supplemental figure in a classical study (Bianchi et al., Nature 2014, 508, 483, Extended figure 8) presenting only one image panel with a few cells that appear to be unfused. To mediate fusion, IZUMO1 must be present on only one of the two membranes suggesting a virus-like unilateral fusion mechanism rather than bilateral mechanisms utilized by *C. elegans* cell fusogen EFF1. Fusogenic activity of IZUMO1 is a very unexpected finding because this protein does not have any obvious homology to known fusogens. It is also a very important finding suggesting that, in analogy to many viral fusogens (for instance, influenza virus hemagglutinin) IZUMO1 mediates both binding and fusion of the membranes. Based on the analysis of a series of mutants the work identifies IZUMO1 domains important for its correct trafficking, cell-cell binding, and cell-cell fusion functions. I am very impressed by the finding that a mutation in the hinge region that is essential for IZUMO1 interaction with JUNO at the surface of oocytes and abolishes binding does not suppress fusion (Fig. 5). In contrast FWW mutant mediates binding but not fusion. Intriguingly, while IZUMO1 fuses BHK cells, it binds but do not fuse BHK cells with oocytes suggesting that fusogenic activity of IZUMO1 in sperm depends on additional factors. The work is well-controlled, elegantly, and clearly written and will be of great interest to many researchers working on fertilization and other cell-cell fusion processes.

Specific comments.

1) Finding that IZUMO1 fuses BHK cells to other BHK cells but not to bound oocytes is very intriguing, and as noted by the authors, indicates that in the context of sperm-egg fusion fusogenic activity of IZUMO1 is controlled by additional factors. I think this requires additional discussion. Can this finding reflect specific properties of cell-cell contacts established by BHK cells (Erickson, 1978, Cell Sci. 33, 53-84)? Very important finding that fusogenic activity of IZUMO1 is not limited to BHK cells and is also observed for HEK293T cells (Fig. S3) can be further strengthened by quantification. Furthermore, a variety of cells, including BHK cells and HEK-293T cells, establish actin-rich nanotube connections between cells (Panasiuk et al., J Virol. 2018 92(10): e00090-18). Can the ability of IZUMO1 to mediate fusion between BHK cells and between HEK293 cells but not between BHK cells and oocytes depend on the presence of the nanotubes with IZUMO1, in effect, driving the expansion of the nanotubes yielding multinucleated cells allowing efficient content mixing.

2) Line 185: In live imaging experiments fusion events "often occur between sister cells several hours after cytokinesis." Please comment on why sister cells can be more likely to fuse. Can it be that sister cells remained connected by a narrow tether and detected fusion events represent enlargement of these connections?

3) Line 349: "In addition, the requirement of a tight binding step to JUNO prior to fusion determines that the sperm cell will fuse to oocytes expressing a species-matching IZUMO1 receptor (Bianchi et al., 2014; Bianchi and Wright, 2015). This is a common mechanism used by many viruses to regulate their cellular tropism even when carrying powerful unilateral fusogens, such as the requirement of CD4 and other co-receptors for HIV infection mediated by the Env glycoprotein (White et al., 2008; Melikyan, 2011)." If by this analogy you mean that BHK cells carry some unidentified receptors activating IZUMO1 for fusion (say by triggering its dimerization) and oocytes do not, this seems counterintuitive.

Minor comments

Line 323: "Proteins related to IZUMO1, for instance, IZUMO2-4, SPACA6 or TMEM95 may work as fusogens or potentiate IZUMO1's fusogenic activity (Ellerman et al., 2009; Deneke and

Pauli, 2021). Notably, IZUMO1 can homo- and hetero-oligomerize (Ellerman et al., 2009; Gaikwad et al., 2019). The absence of these proteins ..." The sentence "Notably, IZUMO1 can homo- and hetero-oligomerize (Ellerman et al., 2009; Gaikwad et al., 2019)." seems to break the discussion of additional proteins that may be involved.

Reviewer #3 (Comments to the Authors (Required)):

This manuscript by Pobjedewicz and colleagues makes the interesting report that the sperm surface factor IZUMO1, which is well characterized to serve as adhesion factor by binding the JUNO protein on the surface of oocytes, also promotes membrane merging in a unilateral manner. The authors discovered the proposed role of IZUMO1 as fusogen by expression in heterologous Baby Hamster Kidney (BHK) cells, where it appears to be sufficient to promote cell-cell fusion in a unilateral manner (expressed only in one of the two fusing cells). They further describe separation-of-function mutations in IZUMO1, which either block JUNO binding and adhesion, or block membrane merging. A large family of virus-derived fusogens is now well established to promote cell-cell membrane fusion in a vast array of organisms but does not appear to be present in mammals. This report would be the first of a bona fide gamete fusogenic protein in mammals, and thus of wide interest.

The report is well presented and reads very easily. However, I am yet to be fully convinced that the range of assays shown to support the view that IZUMO1 functions as a fusogen, as each have some issues. The most convincing assay is the content-mixing assay, where cells of different colors are mixed and dual-color cells counted, as this cannot result from division defects.

The multinucleation assay is much less convincing. First, in figure 1, the % of syncytia is high, at about 25%, but the negative control also shows about 10%, which is quite high. In the methods explaining how multinucleation percentage were determined, could you specify what is counted in the total number nuclei, in particular what is meant by "cells that were in contact" (line 471)? Are these all green cells and all the cells they contact, or only contacting green cells (two green cells or more in contact). Given IZUMO1 is proposed to function in a unilateral manner, the first counting method would be more appropriate. I am a bit concerned that the second method was chosen, which would overestimate the number of fusion events.

Second, syncytia can result from cell fusion or from division defects. Indeed, in the live examination, the text specifies that the majority of cell fusions (which is very low at just over 2%) were between sister cells. To my eyes, the examples shown in figure 3 look like failed cytokinesis events, where the cells may have abscission defects. How can you be sure that the sister cells were fully separated before re-fusion? What is the percentage of cells exhibiting fusion between non-sister cells? Providing this number would be necessary to ascertain that the formation of the binucleate cells is really the result of a fusion event. Ideally, the live-cell experiment should be done with two-color cells, to catch fusion events between (unrelated) cells of different color, or at least show examples of fusion between non-sister cells.

All experiments above are with BHK cells. Figure S3B, which is described to show the quantification of HEK293T cell fusion, is missing. Because the numbers are not stated in the text, it is not possible to evaluate this experiment at this stage.

Finally, the observation that IZUMO is unable to promote fusion of heterologous cells to oocytes - yet does promote adhesion in a JUNO-dependent manner, indicating that there is very close apposition of the cell and oocyte plasma membranes - also does not encourage that IZUMO's central function is as a fusogen. The authors propose possible explanations in their discussion, but it seems a little difficult to imagine that cofactors are missing in the BHK cells to fuse with the oocyte, but not with other BHK cells that should be non-fusogenic.

Reviewer #1 (Comments to the Authors (Required)):

Considerable work has implicated the sperm protein Izumo and its egg receptor, Juno, in mammalian sperm-egg fusion. Work has also documented localization dynamics and structural transitions in Izumo that precede fusion (Inoue, 2015; see model in Fig. 6, Inoue, 2015). However, prior studies have shown that while somatic cells expressing Izumo bound to somatic cells expressing Juno, they did not fuse (Bianchi 2014; Inoue, 2015); similarly somatic cells expressing Izumo bound to, but did not fuse with, mammalian eggs (Inoue 2015). Even cells expressing Izumo plus four other sperm factors, or Izumo plus DCST1 and DSCT2, failed to fuse with oocytes (see Noda 2022).

The current study demonstrates fusion activity for Izumo for the first time. BHK cells expressing Izumo fuse to BHK cells expressing Juno (and to mock transfected BHK cells!), as shown in both syncytia (Fig. 1), and content mixing (Figs. 2 and 3) assays. The authors also provide mutational evidence for a role for three exposed aromatic residues for BHK-Izumo-to-BHK-Juno fusion (Fig. 5: content mixing assay). But, as shown previously (e.g., Inoue 2015), BHK-Izumo cells bind to, but do not fuse with, oocytes (Fig. S4). So, this study demonstrates the fusion activity of Izumo for the first time, but how this happens mechanistically in the context of sperm and eggs remains to be elucidated.

Main Comment

While mutation of 3 aromatic residues (F28, W88, and W113) diminishes the fusion activity of Izumo, these residues do not appear (from Fig. 4A) to form any type of fusion peptide, fusion loop, or fusion (hydrophobic) patch* (as stated on Lines 276-277 in the Discussion) that might mediate protein-lipid interactions.

(a) Perhaps a space-filling model colored on a hydrophobicity scale would be informative as part of Fig. 4.

Fig. 4 has been modified following the Reviewer's suggestion (see Figure 4B; Lines 1026-1029). Additionally, Movie S7 was added for a better perspective on the matter.

(b) Also, please make this point in the Results section when discussing the triple mutant.

A sentence was included in Results (Lines 260-262).

(c) It is suggested to edit the sentence on Lines 375-377 of the Discussion. The author's model (Fig. S5) is quite in line with the model proposed in Fig. 6 of Inoue 2015, albeit without proposing a second oocyte receptor for Izumo.

This sentence was modified accordingly (Lines 384-385) and a reference to a recent new IZUMO1's partner was also added. It is also mentioned in the Legend of Figure S3E (previously S5).

- (d) The work would be stronger if there were insight into what the three aromatic residues do to promote fusion: e.g., testing some of the proposals raised on Lines 291-295 of the Discussion.

This point is extremely interesting and will be addressed in future investigations, being out of the scope of the current Report.

[*All fusexins (viral and cellular), as well as the reovirus FAST proteins, contains a 'coherent' hydrophobic region (fusion peptide or fusion loop) that has (or has been strongly implicated) to mediate the fusogen-lipid bilayer interaction that precedes fusion.]

Minor Comments

1. Line 158: This could be stated more strongly. All tested viral fusion proteins (and the related syncytins) can induce fusion when expressed in only one cell (donor cell) of a set of cells. This is also true for the reovirus FAST proteins.

This line was accordingly modified (Line 157-158).

2. Reovirus FAST proteins seem to warrant a mention in this manuscript, as they are structurally distinct from fusexins and Izumo, but are still sufficient to mediate fusion (i.e., are fusogens).

We completely agree with the reviewer and FAST proteins are now mentioned as examples of unilateral fusogens (Lines 157-158)

3. Fig. 3A legend. Clarify/add arrow(head)s. The reader does not see 'two individual fusion events'.

The legend was edited. Indeed, only one fusion event is shown.

4. Line 194: Suggested edit: 'unilateral mechanism to fuse somatic mammalian cells'.

We appreciate the suggestion and modified the sentence (Lines 194-195).

5. Figs. S2 and S3 appear to be the same. The legends say that S2 is with BHK cells and S3 is with 293T cells.

We appreciate the comment and apologize for this. During the submission process Figure S2 was duplicated. We have corrected this.

6. Results section (especially re: Fig. 5): Please further specify which binding and fusion assays were employed for each statement (e.g., BHK-oocyte binding; BHK-BHK content mixing fusion; BHK-oocyte fusion).

This is clarified in the revised manuscript.

7. Line 241: You might add: 'as is the case for viral fusion proteins' (and FFs if tested).

This addition is included in the current version of the manuscript (Lines 245-246).

8. Fig. S4: It is suggested to provide quantitation in the figure proper (currently now in legend). You could put % fused (of bound) cells in the picture boxes or provide a new panel with the quantitation.

The % of fusion was added in the figure (now Figure S3). There is also a detailed explanation of the results in the legend.

9. Fig. S4 legend: Mention that cell complexes were exposed to low (?what) pH to activate VSV G? Were samples in all panels of Fig. S4 exposed to low pH (e.g., ones in top panels, Izumo alone)? This wasn't clear. Relatedly, does low pH treatment inactivate the fusion activity of Izumo? If this hasn't been assessed, can you test that with one of your BHK Izumo fusion assays? While not deemed highly likely, it would be good to rule out this possibility.

The experiment showed for IZUMO1 alone was performed at neutral pH. This is better clarified in the legend of Figure S3 (Previous S4). Nevertheless, in preliminary assays we tried lowering the pH also for IZUMO1 without seeing fusion. The pH employed was mentioned previously in the Material and Methods section, however, now is included also in the legend of the figure.

10. Line 218: Provide a reference (and PDB #) for the Izumo crystal structure.

This information was added accordingly (Lines 219-220).

11. Line 227: Suggested edit: 'like certain members of the Fusexin family'. (Not all viral fusion proteins have IgG domains.)

We appreciate the comment. Indeed, not all viral fusion proteins have IgG domains, however, all members of the Fusexin family do. We have modified the sentence to make it clearer (Lines 226-229).

12. Line 273: Specify 'heterologous somatic cell-to-cell fusion' (or 'BHK cell-to-cell fusion'), since BHK-Izumo cells did not fuse with oocytes.

This sentence was modified according to the Reviewer's suggestion (Line 269).

13. Line 279: Clarify/elaborate: Which fusexins do not have a fusion peptide or fusion loop? How do they function in fusion without such an element?

We now elaborate on alternative models of how EFF-1 may mediate membrane fusion as an example of a fusogen without any hydrophobic loop or patch. We also refer to a recent Review article (Lines 283-289).

Reviewer #2 (Comments to the Authors (Required)):

Sperm protein IZUMO1 is essential for sperm-egg binding in mammalian fertilization. This binding is followed by cell fusion and the identity of the protein(s) that fuse the plasma membranes of sperm and egg has remained one of the most important open questions in the field of fertilization. Brukman et al. show that expression of IZUMO1 in mammalian cells that do not normally fuse is sufficient to induce their fusion detected as formation of multinucleated cells and content mixing. This work corrects an earlier belief in the field that IZUMO1 does not fuse cells based on just one supplemental figure in a classical study (Bianchi et al., Nature 2014, 508, 483, Extended figure 8) presenting only one image panel with a few cells that appear to be unfused. To mediate fusion, IZUMO1 must be present on only one of the two membranes suggesting a virus-like unilateral fusion mechanism rather than bilateral mechanisms utilized by *C. elegans* cell fusogen EFF1. Fusogenic activity of IZUMO1 is a very unexpected finding because this protein does not have any obvious homology to known fusogens. It is also a very important finding suggesting that, in analogy to many viral fusogens (for instance, influenza virus hemagglutinin) IZUMO1 mediates both binding and fusion of the membranes. Based on the analysis of a series of mutants the work identifies IZUMO1 domains important for its correct trafficking, cell-cell binding, and cell-cell fusion functions. I am very impressed by the finding that a mutation in the hinge region that is essential for IZUMO1 interaction with JUNO at the surface of oocytes and abolishes binding does not suppress fusion (Fig. 5). In contrast FWW mutant mediates binding but not fusion. Intriguingly, while IZUMO1 fuses BHK cells, it binds but do not fuse BHK cells with oocytes suggesting that fusogenic activity of IZUMO1 in sperm depends on additional factors. The work is well-controlled, elegantly, and clearly written and will be of great interest to many researchers working on fertilization and other cell-cell fusion processes.

Specific comments.

1) Finding that IZUMO1 fuses BHK cells to other BHK cells but not to bound oocytes is very intriguing, and as noted by the authors, indicates that in the context of sperm-egg fusion fusogenic activity of IZUMO1 is controlled by additional factors. I think this requires additional discussion. Can this finding reflect specific properties of cell-cell contacts established by BHK cells (Erickson, 1978, Cell Sci. 33, 53-84)? Very important finding that fusogenic activity of IZUMO1 is not limited to BHK cells and is also observed for HEK293T cells (Fig. S3) can be further strengthened by quantification. Furthermore, a variety of cells, including BHK cells and HEK-293T cells, establish actin-rich nanotube connections between cells (Panasiuk et al., J Virol. 2018 92(10): e00090-18). Can the ability of IZUMO1 to mediate fusion between BHK cells and between HEK293 cells but not between BHK cells and oocytes depend on the presence of the nanotubes with IZUMO1, in effect, driving the expansion of the nanotubes yielding multinucleated cells allowing efficient content mixing.

Figure S2B and C shows the results for HEK293T including the quantitation. The hypothesis involving nanotubes is indeed very interesting. Even though IZUMO1 could promote the expansion of nanotubes, the increased content mixing can only be explained by cell-cell fusion or the formation *de novo* of more nanotubes that in essence requires membrane fusion. We now discuss this hypothesis in the manuscript (Lines 336-340).

2) Line 185: In live imaging experiments fusion events "often occur between sister cells several hours after cytokinesis." Please comment on why sister cells can be more likely to fuse. Can it be that sister cells remained connected by a narrow tether and detected fusion events represent enlargement of these connections?

This phenomenon of fusion after cytokinesis was already described in mammalian and non-mammalian cells (e.g. Yu et al., 2015 DOI: 10.1021/tx500473h, Chan et al., 2022 DOI: 10.1038/s41586-022-04641-0). First, after cell division cells are in intimate contact which may favor cell fusion. Secondly, other common factor like cell-cycle stage may also influence this observation. In any case, the fusion between non-related cells can be confirmed by the appearance of mixed cells in the content mixing experiments which complements the live-imaging results. We now comment on this in the manuscript (Lines 185-186). In addition we added an experiment with FdUrd that blocks the cell cycle and has no effect on multinucleation supporting the interpretation that syncytia formation mediated by IZUMO1 is not due to a failure in cytokinesis but to cell fusion (Figure S1D; Lines 136-141).

3) Line 349: "In addition, the requirement of a tight binding step to JUNO prior to fusion determines that the sperm cell will fuse to oocytes expressing a species-matching IZUMO1 receptor (Bianchi et al., 2014; Bianchi and Wright, 2015). This is a common mechanism used by many viruses to regulate their cellular tropism even when carrying powerful unilateral fusogens, such as the requirement of CD4 and other co-receptors for HIV infection mediated by the Env glycoprotein (White et al., 2008; Melikyan, 2011). " If by this analogy you mean that BHK cells carry some unidentified receptors activating IZUMO1 for fusion (say by triggering its dimerization) and oocytes do not, this seems counterintuitive.

A tight binding between two fusing cells is a pre-requisite for fusion (Hernandez and Podbilewicz 2017, DOI: 10.1242/dev.155523). BHK cells in culture show homotypic interactions mediated by the specific action of adhesion molecules like N-CAM (Thoulouze et al., 1998. DOI: 10.1128/jvi.72.9.7181-7190.1998). In the light of our results, it is possible that the combined activity of these molecules is sufficient to by-pass the absence of JUNO in somatic cell-to-cell fusion in vitro. In the case of the sperm, the IZUMO1-JUNO interaction is required for the tight binding of the sperm to the oocyte in a cell-type specific manner giving a tropism to IZUMO1 fusogenic activity. This could explain why we could only detect sperm fusion to somatic cells expressing JUNO (Fig. S3A and B).

Minor comments

Line 323: "Proteins related to IZUMO1, for instance, IZUMO2-4, SPACA6 or TMEM95 may work as fusogens or potentiate IZUMO1's fusogenic activity (Ellerman et al., 2009; Deneke and Pauli, 2021). Notably, IZUMO1 can homo- and hetero-oligomerize (Ellerman et al., 2009; Gaikwad et al., 2019). The absence of these proteins ..." The sentence "Notably, IZUMO1 can homo- and hetero-oligomerize (Ellerman et al., 2009; Gaikwad et al., 2019)." seems to break the discussion of additional proteins that may be involved.

This sentence was modified in the revised manuscript (Lines 329-330).

Reviewer #3 (Comments to the Authors (Required)):

This manuscript by Pobjedewicz and colleagues makes the interesting report that the sperm surface factor IZUMO1, which is well characterized to serve as adhesion factor by binding the JUNO protein on the surface of oocytes, also promotes membrane merging in a unilateral manner. The authors discovered the proposed role of IZUMO1 as fusogen by expression in heterologous Baby Hamster Kidney (BHK) cells, where it appears to be sufficient to promote cell-cell fusion in a unilateral manner (expressed only in one of the two fusing cells). They further describe separation-of-function mutations in IZUMO1, which either block JUNO binding and adhesion, or block membrane merging. A large family of virus-derived fusogens is now well established to promote cell-cell membrane fusion in a vast array of organisms but does not appear to be present in mammals. This report would be the first of a bona fide gamete fusogenic protein in mammals, and thus of wide interest.

The report is well presented and reads very easily. However, I am yet to be fully convinced that the range of assays shown to support the view that IZUMO1 functions as a fusogen, as each have some issues. The most convincing assay is the content-mixing assay, where cells of different colors are mixed and dual-color cells counted, as this cannot result from division defects.

The multinucleation assay is much less convincing. First, in figure 1, the % of syncytia is high, at about 25%, but the negative control also shows about 10%, which is quite high. In the methods explaining how multinucleation percentage were determined, could you specify what is counted in the total number nuclei, in particular what is meant by "cells that were in contact" (line 471)? Are these all green cells and all the cells they contact, or only contacting green cells (two green cells or more in contact). Given IZUMO1 is proposed to function in a unilateral manner, the first counting method would be more appropriate. I am a bit concerned that the second method was chosen, which would overestimate the number of fusion events.

We appreciate your comments. We have now specified the total number of nuclei counted as expressing cells in contact (Methods Lines 478-481 and legend for Figure 1B; Lines 968-971). We use this assay for initial screening for fusogenic activities of different proteins. Because a priori we do not know if a fusogen works in a bilateral or unilateral mechanism we assume bilaterality, analyzing transfected cells in contact, and when using content mixing assays, we discriminate unilateral/bilateral mechanisms. Since the report shows the sequential steps that led us to discover the unexpected fusogenic activity of IZUMO1, we would like to keep the figure 1 in the main text.

Second, syncytia can result from cell fusion or from division defects.

To rule out the possibility that division defects may contribute to multinucleation, we have added a new control in Figure S1D: “To determine whether IZUMO1-induced syncytia formation was due to failure in cytokinesis, we incubated BHK cells with and without an inhibitor of the cell cycle (5-fluoro-2'-deoxyuridine; Valansi et al., 2017). We found the cells failed to proliferate but there was no reduction in multinucleation, suggesting that IZUMO1-mediated multinucleation is not due to failure in cytokinesis” (Lines 136-141).

Indeed, in the live examination, the text specifies that the majority of cell fusions (which is very low at just over 2%) were between sister cells. To my eyes, the examples shown in figure 3 look like failed cytokinesis events, where the cells may have abscission defects. How can you be sure that the sister cells were fully separated before re-fusion? What is the percentage of cells exhibiting fusion between non-sister cells? Providing this number would be necessary to ascertain that the formation of the binucleate cells is really the result of a fusion event. Ideally, the live-cell experiment should be done with two-color cells, to catch fusion events between (unrelated) cells of different color, or at least show examples of fusion between non-sister cells.

The defects in cell division would be reversible at relatively short times, here we can see fusion several hours after division. In fact, in the fusion showed for IZUMO1 it can be seen in the Z-stack (Movie S6) that the cells are clearly separated at the resolution of light microscopy. Furthermore, as commented by the Reviewer, the content mixing assay complements this technique excluding cell division defects. The shortest window of analysis during live-imaging vs. content mixing could explain why fusion events are more difficult to record in time-lapse microscopy. We did observe examples of fusion which involves two cells that did not result from a division, however, this fusion is harder to see and therefore we decided to keep the one that is included currently (See Figure below and attached movie).

IZUMO1-Venus

All experiments above are with BHK cells. Figure S3B, which is described to show the quantification of HEK293T cell fusion, is missing. Because the numbers are not stated in the text, it is not possible to evaluate this experiment at this stage.

We are afraid that the Figure S3 uploaded with the original manuscript was a duplication of Figure S2. We have now updated the new Figure S2 which includes the quantification (Figure S2C). We apologize for the inconvenience and thank the Reviewer for noting this issue.

Finally, the observation that IZUMO is unable to promote fusion of heterologous cells to oocytes - yet does promote adhesion in a JUNO-dependent manner, indicating that there is very close apposition of the cell and oocyte plasma membranes - also does not encourage that IZUMO's central function is as a fusogen. The authors propose possible explanations in their discussion, but it seems a little difficult to imagine that cofactors are missing in the BHK cells to fuse with the oocyte, but not with other BHK cells that should be non-fusogenic.

We appreciate the Reviewer's comment. It is our intention to explore the different possibilities in future experiments including PS exposure, mechanical forces and the co-expression of different sperm factors.

November 2, 2022

RE: JCB Manuscript #202207147R

Prof. Benjamin Podbilewicz
Technion - Israel Institute of Technology
Biology
Technion City
Haifa 32000
Israel

Dear Prof. Podbilewicz:

Thank you for submitting your revised manuscript entitled "A novel function for the sperm adhesion molecule IZUMO1 as a cell-cell fusion protein". We would be happy to publish your paper in JCB pending final revisions necessary to meet our formatting guidelines (see details below) as well as the two final requests made by Reviewer 3.

A. MANUSCRIPT ORGANIZATION AND FORMATTING:

Full guidelines are available on our Instructions for Authors page, <http://jcb.rupress.org/submission-guidelines#revised>. Submission of a paper that does not conform to JCB guidelines will delay the acceptance of your manuscript.

1) Text limits: Character count for Reports is < 20,000 characters, not including spaces. Count includes abstract, introduction, results, discussion, and acknowledgments. Count does not include title page, figure legends, materials and methods, references, tables, or supplemental legends.

2) Figures limits: Reports may have up to 5 main text figures and 3 supplemental figures.

3) Figure formatting: Scale bars must be present on all microscopy images, including inset magnifications. Molecular weight or nucleic acid size markers must be included on all gel electrophoresis.

4) Statistical analysis: Error bars on graphic representations of numerical data must be clearly described in the figure legend. The number of independent data points (n) represented in a graph must be indicated in the legend. Statistical methods should be explained in full in the materials and methods. For figures presenting pooled data the statistical measure should be defined in the figure legends. Please also be sure to indicate the statistical tests used in each of your experiments (either in the figure legend itself or in a separate methods section) as well as the parameters of the test (for example, if you ran a t-test, please indicate if it was one- or two-sided, etc.). Also, if you used parametric tests, please indicate if the data distribution was tested for normality (and if so, how). If not, you must state something to the effect that "Data distribution was assumed to be normal but this was not formally tested."

**Please indicate the n for each figure panel in the appropriate figure legend and not in Table S1.

5) Abstract and title: The abstract should be no longer than 160 words and should communicate the significance of the paper for a general audience. The title should be less than 100 characters including spaces. Make the title concise but accessible to a general readership.

** While your current title is very good, please consider a slightly more concise title such as:

"A novel function for the sperm adhesion protein IZUMO1 in cell-cell fusion"

Please let us know if you have other ideas for the title.

6) Materials and methods: Should be comprehensive and not simply reference a previous publication for details on how an experiment was performed. Please provide full descriptions in the text for readers who may not have access to referenced manuscripts.

7) Please be sure to provide the sequences for all of your primers/oligos and RNAi constructs in the materials and methods. You must also indicate in the methods the source, species, and catalog numbers (where appropriate) for all of your antibodies. Please also indicate the acquisition and quantification methods for immunoblotting/western blots.

8) Microscope image acquisition: The following information must be provided about the acquisition and processing of images:

a. Make and model of microscope

b. Type, magnification, and numerical aperture of the objective lenses

- c. Temperature
- d. Imaging medium
- e. Fluorochromes
- f. Camera make and model
- g. Acquisition software
- h. Any software used for image processing subsequent to data acquisition. Please include details and types of operations involved (e.g., type of deconvolution, 3D reconstitutions, surface or volume rendering, gamma adjustments, etc.).

10) Supplemental materials: There are strict limits on the allowable amount of supplemental data. Articles may have up to 5 supplemental figures. Please also note that tables, like figures, should be provided as individual, editable files. A summary of all supplemental material should appear at the end of the Materials and methods section.

13) ORCID IDs: ORCID IDs are unique identifiers allowing researchers to create a record of their various scholarly contributions in a single place. At resubmission of your final files, please consider providing an ORCID ID for as many contributing authors as possible.

14) A separate author contribution section following the Acknowledgments. All authors should be mentioned and designated by their full names. We encourage use of the CRedit nomenclature.

Please note that JCB now requires authors to submit Source Data used to generate figures containing gels and Western blots with all revised manuscripts. This Source Data consists of fully uncropped and unprocessed images for each gel/blot displayed in the main and supplemental figures. Since your paper includes cropped gel and/or blot images, please be sure to provide one Source Data file for each figure that contains gels and/or blots along with your revised manuscript files. File names for Source Data figures should be alphanumeric without any spaces or special characters (i.e., SourceDataF#, where F# refers to the associated main figure number or SourceDataFS# for those associated with Supplementary figures). The lanes of the gels/blots should be labeled as they are in the associated figure, the place where cropping was applied should be marked (with a box), and molecular weight/size standards should be labeled wherever possible.

WHEN APPROPRIATE: The source code for all custom computational methods published in JCB must be made freely available as supplemental material hosted at www.jcb.org. Please contact the JCB Editorial Office to find out how to submit your custom macros, code for custom algorithms, etc. Generally, these are provided as raw code in a .txt file or as other file types in a .zip file. Please also include a one-sentence summary of each file in the Online Supplemental Material paragraph of your manuscript.

B. FINAL FILES:

-- Cover images: If you have any striking images related to this story, we would be happy to consider them for inclusion on the journal cover. Submitted images may also be chosen for highlighting on the journal table of contents or JCB homepage carousel.

Images should be uploaded as TIFF or EPS files and must be at least 300 dpi resolution.

****It is JCB policy that if requested, original data images must be made available to the editors. Failure to provide original images upon request will result in unavoidable delays in publication. Please ensure that you have access to all original data images prior to final submission.****

****The license to publish form must be signed before your manuscript can be sent to production. A link to the electronic license to publish form will be sent to the corresponding author only. Please take a moment to check your funder requirements before choosing the appropriate license.****

Thank you for this interesting contribution, we look forward to publishing your paper in Journal of Cell Biology.

Sincerely,

Michael Rudnicki
Monitoring Editor
Journal of Cell Biology

Tim Fessenden
Scientific Editor
Journal of Cell Biology

Reviewer #3 (Comments to the Authors (Required)):

The authors have answers most of my concerns, but have not provided quantification of the % of fusion between non-sister cells. The example they provide in the response to reviewers is also not completely convincing that the two cells that fuse are not also sister cells. I agree that there may be good reasons for fusion to occur preferentially between sister cells, and they have done other controls which reinforce this point. However, I would ask that they modify the text in line 190 to remove "often occur" and replace by either the actual % of cases or something like "typically occur".

The second minor point I ask is to remove the word "crucially" from the last sentence of the abstract. The paper does not show whether the fusogenic function of IZUMO is crucial where it is normally physiologically expressed during fertilization. In fact, the experiments aiming to fuse somatic cells with oocytes failed.